# REMO2020: a modernized modular regional climate model

Joni-Pekka Pietikäinen<sup>1</sup>, Kevin Sieck<sup>1</sup>, Lars Buntemeyer<sup>1</sup>, Thomas Frisius<sup>1</sup>, Christine Nam<sup>1</sup>, Peter Hoffmann<sup>1</sup>, Christina Pop<sup>1</sup>, Diana Rechid<sup>1</sup>, and Daniela Jacob<sup>1</sup>

<sup>1</sup>Climate Service Center Germany (GERICS), Helmholtz-Zentrum Hereon, Fischertwiete 1, 20095 Hamburg, Germany Correspondence: Joni-Pekka Pietikäinen (Joni-Pekka.Pietikaeinen@hereon.de)

### Abstract.

This paper introduces REMO2020, a modernized version of the well-known and widely used REMO regional climate model. REMO2020 has undergone fundamental changes in its code structure to provide a more modular, operationally focused design, facilitating the inclusion of new components and updates. Here, we describe the default configuration of REMO2020, which includes the following updates compared to the previous version, REMO2015: (i) the FLake lake model; (ii) a state-of-the-art MACv2-SP aerosol climatology; (iii) a newly developed 3-layer snow module; (iv) a prognostic precipitation scheme; (v) an updated time filter; and (vi) a new tuning approach. Additionally, we describe some optional modules that can be activated separately, such as the interactive MOsaic based VEgetation model iMOVE. REMO2020 outperforms its predecessor REMO2015 in nearly all evaluation metrics used to evaluate simulations of Europe's climate. The persistent warm temperature bias over Central Europe and cold temperature bias over Northern Europe have been significantly reduced in REMO2020. Similarly, the previously modeled dry bias in Central Europe has been nearly eliminated, and the extent of the wet bias in Eastern Europe has been reduced. The precipitation distribution in REMO2020 is much more realistic, especially in terms of heavy precipitation extremes. Statistically, REMO2020 aligns better with long-term measurements than older versions. Mountainous areas still present a challenge in REMO2020, especially with higher vertical resolution. In this paper, we demonstrate why REMO2020 will be our new model version for future dynamical downscaling activities.

#### 1 Introduction

25

In this paper, we present the new model version REMO2020 of the REgional climate MOdel REMO (Jacob and Podzun, 1997; Jacob, 2001). Previous versions of REMO, a widely used regional climate model (RCM), have been used for basic climate research as well as more complex climate service science supporting societies and political decisions (e.g., IPCC, 2007, 2014, 2019, 2021; Jacob et al., 2020, ?). Over the past decade, REMO has participated in the World Climate Research Programme (WCRP)'s Coordinated Regional Climate Downscaling Experiment (CORDEX) in which RCM dynamical downscaling simulations are performed covering almost all land areas of the entire globe (Giorgi et al., 2009). Before this, REMO participated in dynamical downscaling projects such as PRUDENCE (Christensen et al., 2007; Jacob et al., 2007) and ENSEMBLES (Linden and Mitchell, 2009).

Some of the CORDEX domains, like the European EURO-CORDEX domain (Jacob et al., 2014, 2020), have a higher number of participating modelling centers. Similarly, some CORDEX domains have focused on higher resolution approaches,

e.g., the EURO-CORDEX domain is defined at 0.11° in addition to the 0.44° standard resolution of CORDEX. CORDEX also provides the CORDEX-COmmon Regional Experiment (CORE) Framework (Gutowski Jr. et al., 2016), which produces more homogeneous high-resolution regional climate information covering almost the whole globe (Remedio et al., 2019; Ciarlo et al., 2020; Teichmann et al., 2020; Coppola et al., 2021). These resolutions, of ~10-20 km, allows RCMs to better represent climate extremes, such as heavy precipitation, compared to general circulation models (GCMs) in long transient climate simulations (Rummukainen, 2016; Goergen and Kollet, 2021; Kotlarski et al., 2014; Haarsma et al., 2016).

In 2016, CORDEX launched the first call for "Flagship Pilot Studies (FPS)" with targeted experimental setups to better address key scientific questions motivated by a number of challenges such as downscaling to convection permitting scales (e.g., Coppola et al., 2020), followed by investigating regional scale forcing like aerosols and land use changes (e.g., Davin et al., 2020), or urban environments (Halenka and Langendijk, 2022; Langendijk et al., 2024). To answer the FPS questions, the models require targeted developmental improvements arising from scientific needs including higher-resolutions and longer-term climate simulation requirements, determining the level of detail and model complexity. Moreover, the input/forcing data used by the models needs to be constantly updated and the models need to be ready for such datasets, highlighting again the development needs (Hoffmann et al., 2023; Katragkou et al., 2024; Langendijk et al., 2024). Recently, high-resolution non-hydrostatic kilometer-scale simulations have been performed with RCMs for longer time periods (decade-scales) to even further improve the representation of extremes (Coppola et al., 2020; Pichelli et al., 2021; Ban et al., 2021; Fosser et al., 2024). Model development has been a key aspect in achieving these steps.

Additionally, we are also moving towards regional Earth System Models (ESMs), which require up-to-date components, such as lakes, vegetation/land-use, mesoscale atmosphere, ocean circulation features, aerosols etc. (Giorgi and Gao, 2018). Moreover, the existing approaches in the models should be frequently evaluated to identify issues with the components or the utilization approaches. This was pointed out by a study from Boé et al. (2020), in which the authors showed that underestimating the need for time-varying anthropogenic aerosols can limit the RCM ensemble ability to capture the upper part of the climate change uncertainty range. Missing or overly simplified components of the earth system can cause significant biases in the results (Kotlarski et al., 2014; Pietikäinen et al., 2018). Thus, further new developments and updates to existing components are crucial, especially now that models are used in convection permitting scales (Giorgi et al., 2023).

The changes in REMO2020 presented here focus mainly on the model physics, though we have also made dynamical and structural changes. Our simulations are done with the hydrostatic dynamics, and the recent non-hydrostatic model development steps will be presented in separate future studies. This article is structured as follows: Section 2 presents all the new developments and updates in detail, followed by Section 3, which describes our simulation setup and analysis/observational data. Section 4 shows and discusses the model evaluation and the actual data analysis, and finally, Section 5 presents the main conclusions.

#### 2 Model and developments

65

85

In this Section the regional climate model REMO is introduced. We begin with a history of the model, followed by an overview of its main components. Particular emphasis is placed on the land surface scheme, unique to REMO, which has not been previously presented in such detail. First, we describe the previous REMO version, REMO2015, followed by the advancements to various physical packages in the REMO2020 version. In addition, the contributions of REMO2020 to the wider WCRP-CORDEX activities will also be presented. Since REMO2015, the model has also been capable of running in non-hydrostatic mode, which also has been vastly improved with REMO2020. In this study, we focus on the hydrostatic part and leave the details of non-hydrostatic tuning and set-up for future studies.

## 2.1 REgional climate MOdel REMO

REMO is a three-dimensional limited-area atmosphere model originally developed at the Max Planck Institute for Meteorology in Hamburg, Germany, and currently further developed at the Climate Service Center Germany (GERICS) in Hamburg, an organisation of the Helmholtz-Zentrum Hereon. It has been successfully used since 1997 in various international projects including CORDEX (Jacob and Podzun, 1997; Jacob, 2001; Jacob et al., 2001, 2007; Teichmann, 2010; Lorenz and Jacob, 2014; Remedio et al., 2019; Coppola et al., 2020; Davin et al., 2020; Giorgi et al., 2022). Historically, the model core's roots are in the Europa Model (EM), the former numerical weather prediction (NWP) model of the German Weather Service (DWD), while the physical packages originated from the global GCM ECHAM4 (Roeckner et al., 1996). Both the dynamical core and physics packages of the model have been frequently revised and updated over the last 20 years (e.g., Hagemann, 2002; Semmler et al., 2004; Pfeifer, 2006; Rechid and Jacob, 2006; Rechid, 2009; Pietikäinen et al., 2012; Preuschmann, 2012; Pietikäinen et al., 2014; Wilhelm et al., 2014; Pietikäinen et al., 2018)). Some updates were made directly to the model's main code, while others have created separate branches.

The prognostic variables in REMO are horizontal wind components, surface pressure, air temperature, specific humidity, cloud liquid water, and cloud ice. Vertical representation in REMO is based on a terrain-following hybrid sigma-pressure coordinate system. A spherical Arakawa-C grid (Arakawa and Lamb, 1977) is used horizontally. In this grid, all prognostic variables, except winds, are defined at the center of a grid box, whereas the wind components are defined at the edges of the grid boxes. Temporal discretization is done by using a leap-frog scheme with time filtering by Asselin (1972). To enable longer time steps, a semi-implicit correction is used. A relaxation scheme by Davies (1976) is used for prognostic variables at the eight outermost grid boxes.

Clouds in REMO are separated into two approaches. The large-scale stratiform cloud scheme is based on the ECHAM5 cloud scheme (Roeckner et al., 1996; Pfeifer, 2006). It includes prognostic equations for cloud water, water vapor, and cloud ice, and utilizes an empirical cloud cover scheme by Sundqvist et al. (1989). The cloud droplet concentration is a height-dependent parameterization and differs for continental and maritime climates (Roeckner et al., 1996). The second cloud approach is the convective (sub-grid) cloud parameterization. Its roots are in the mass-flux scheme from Tiedtke (1989) with modifications by

Nordeng (1994). The scheme also includes some REMO specific modifications, such as the cold convection scheme by Pfeifer (2006), which improves precipitation in cold air outbreaks over oceans in the extra tropics.

The surface in REMO is implemented with a fractional tile approach. Different tile-wise surface schemes and their components have been added or developed into REMO and have been reported in many separate publications, such as (Semmler, 2002; Kotlarski, 2007; Asmus et al., 2023). In the next section, for the first time, the land surface scheme in REMO will be fully explained in detail.

## 2.2 REMO's land surface scheme

100

The land surface scheme of the regional climate model REMO is based on physical parameterizations of the general circulation model ECHAM4 (Roeckner et al., 1996). Over the last couple of decades, it has been improved and expanded upon. Some components include the surface runoff scheme ((Hagemann and Gates, 2003), a sub-grid tile approach (Semmler et al., 2004), inland glaciers (Kotlarski, 2007), vegetation phenology (Rechid and Jacob, 2006; Rechid et al., 2009), interactive MOsaic based VEgetation REMO-iMOVE (Wilhelm et al., 2014), inland lakes and rivers (Pietikäinen et al., 2018), and an irrigation parameterization (Asmus et al., 2023).

The coupling between land and atmosphere is semi-implicit. For vertical diffusion, the turbulent surface fluxes are calculated from Monin-Obukhov similarity theory (Louis, 1979) with a higher-order closure scheme for the transfer coefficients of momentum, heat, moisture, and cloud water within and above the planetary boundary layer. Eddy diffusion coefficients are calculated as functions of the turbulent kinetic energy.

For vertical surface fluxes, the sub-grid tile approach for land, water, and sea ice surfaces was implemented by Semmler et al. (2004). The turbulent surface fluxes and the surface radiation flux are calculated separately for each fraction and are subsequently averaged within the lowest atmospheric level using the respective areas as weights. During the model integration, for each surface tile an individual roughness length, albedo and surface temperature are calculated. The land fraction is further divided into a part covered by vegetation and a bare soil fraction. Over the land fraction, a sub-grid tile for inland glaciers was added by Kotlarski (2007), and for inland lakes and rivers by Pietikäinen et al. (2018). In REMO-iMOVE, a sub-grid tile for irrigated crops was implemented by Asmus et al. (2023).

The land surface parameters, allocated to major ecosystem types according to the classification of Olson (1994a, b), is derived from Hagemann et al. (1999) and Hagemann (2002). The distribution of the Olson ecosystem types was derived from Advanced Very High Resolution Radiometer (AVHRR) data at 1 km resolution, supplied by the International Geosphere-Biosphere Program (Eidenshink and Faundeen, 1994) and constructed by the U.S. Geological Survey (USGS, 2002). For each land cover type, parameter values for the vegetation properties are specified. This information is aggregated to the model grid scale by averaging the vegetation parameters of all land cover types, which are located in one model grid cell. The vegetation cover is represented by parameter values for leaf area index (LAI, ratio of one-sided leaf area to ground area), fraction of photosynthetically active vegetation, background surface albedo (albedo over snow-free land surfaces), surface roughness length of vegetation (integrated with roughness length of topography), fractional forest cover, and water holding capacity.

The seasonal variability of vegetation is represented by monthly varying fields of LAI and fractional green vegetation cover (Rechid and Jacob, 2006). The seasonal variation of the LAI between minimum and maximum values is estimated by a global data field of the monthly growth factor, defined by climatologies of 2 m temperature and the fraction of photosynthetic absorbed radiation (Hagemann, 2002). This dataset is prescribed to the climate model simulations as a mean annual vegetation cycle without interannual variability. In the study by Rechid et al. (2009), an advanced parameterization of the snow-free land surface albedo was developed, describing the seasonal variation of the surface albedo as a function of the monthly varying LAI by using data products from the Moderate-Resolution Imaging Spectroradiometer (MODIS). This provides a basis for the treatment of land surface albedo within a dynamic phenology scheme.

The vegetation parameter values are prescribed as lower boundary conditions and influence the vertical exchange of water and energy at the land surface between the atmosphere and the underlying soil. The surface albedo determines the short-wave radiation budget at the Earth's surface. The density of vegetation cover, represented by the LAI and green vegetation cover, controls the transpiration by the leaf stomatal conductance and the evaporation by the interception of water on the canopy's skin. Evapotranspiration determines the partitioning of the vertical turbulent heat fluxes into latent and sensible heat. The latent and sensible heat fluxes are the main mechanisms to return energy from the surface into the atmosphere. They influence convective processes and the boundary layer structure. These surface processes controlled by vegetation properties impact the near surface atmospheric conditions such as temperature, humidity, and low-level cloudiness.

Soil temperatures are calculated from diffusion equations solved in five discrete layers (0.0 m to -0.065 m, -0.319 m, -1.232 m, -4.134 m, -9.834 m with zero heat flux at the bottom, according to the scheme of Warrilow (1986). The soil thermal conductivity and heat capacity are described as functions of soil water content according to Semmler (2002). For snow, REMO has a one-layer scheme, but it uses an artificial extra top layer for heat conduction when calculating the influence of residual surface energy fluxes on snow (sum of short-wave, long-wave, sensible heat, and latent heat fluxes). The influence of the extra layer is artificially limited to the top 10 cm of the snow pack, and its temperature is used at the interface between the atmosphere and snow. If the snow pack has more than 10 cm of snow, the snow temperature representing the whole pack is interpolated from the artificial top snow layer and the top soil layer temperatures (Semmler, 2002; Kotlarski, 2007).

Soil hydrology comprises three water budget equations for water storage in the soil-related water reservoirs: snow, skin reservoir (water intercepted by vegetation) and soil. The soil water amount is filled in a single soil moisture reservoir by precipitation and snow melt, and depleted by bare soil evaporation from the upper 10 cm. From below, the water can only evaporate via transpiration. For subsurface drainage, rapid and slow drainage are distinguished. Rapid drainage occurs when the soil moisture is more than 90% of the field capacity, whereas slow drainage occurs at values between 5 and 90% of the field capacity. The maximum amount of plant-available water is allocated according to Hagemann (2002), indirectly considering plant root depth. If the soil moisture content reaches saturation, surface runoff occurs. The runoff scheme considers sub-grid scale variations of the field capacity over inhomogeneous terrain (Dümenil and Todini, 1992), and was advanced by Hagemann and Gates (2003) to consider sub-grid scale variations of soil saturation. REMO also contains the option for using a multilayer hydrology scheme according to Hagemann and Stacke (2015), which has been recently applied by Abel (2023).

REMO has been coupled to the interactive MOsaic based VEgetation (iMOVE, Wilhelm et al., 2014). iMOVE is based on selected modules of the land surface model JSBACH (Reick et al., 2013). Within iMOVE, the land surface is represented by plant functional types (PFTs), whose geographic distribution can be derived from different land cover datasets (e.g., Reinhart et al., 2022; Hoffmann et al., 2023). Currently, 14 PFTs (12 vegetation PFTs and 2 crop PFTs) and 2 land surfaces types (i.e., urban and bare ground) are implemented (Wilhelm et al., 2014). The PFT concept includes biophysiological characteristics and functional traits of vegetation, which affect land-atmosphere interactions. iMOVE is fully coupled to REMO at each model time step, enabling plant processes to react to atmospheric and soil conditions, and vice-versa, through land-atmosphere interactions. Plant processes such as photosynthesis, respiration, and transpiration are included in iMOVE, as well as the dependence of their stomatal conductance on atmospheric CO<sub>2</sub> levels driving evapotranspiration. The LAI and the fractional green vegetation cover of a grid cell evolve through plant growth, employing a logistic growth model influenced by temperature-driven growing seasons and crop harvesting. The incorporation of these biophysiological plant processes improves the representation of the vegetation cycle by considering dynamic inter-annual variability.

Within this work, we have changed the calculation of the LAI in iMOVE. The LAI develops during the growing season, which is defined by a temperature-based threshold. The end of the growing season is defined for crops by a harvest event, which decreases the LAI to a minimum. Wilhelm et al. (2014) proposed a fast LAI decrease for harvesting:

$$LAI = LAI * \exp(a * \Delta t_{\text{days}}), \tag{1}$$

where  $\Delta t_{\rm days}$  is the time step length in days and a = -0.1428 [1/day]. But this leads to an unrealistic fast decrease of LAI and reaches its minimum too early in the year. Thus, we integrated a slower LAI decrease for non-tropical areas using a = -0.0333 [1/day]. By default, REMO2020 uses the slower LAI decrease.

Further improvements of iMOVE compared to the standard land surface scheme in REMO include the representation of the background land surface albedo as a combination of soil moisture-dependent bare soil albedo and vegetation albedo, as well as the representation of evaporation of bare soil, which occurs even for soil moisture lower than 90% in iMOVE (Wilhelm et al., 2014). A full documentation and evaluation of the iMOVE module can be found in (Wilhelm et al., 2014). In This work, we want to present more surface-oriented results from the new model version, although the actual evaluation of iMOVE was done by Wilhelm et al. (2014). Moreover, an irrigation parameterisation has been implemented into REMO2020-iMOVE and evaluated by Asmus et al. (2023). REMO-iMOVE has been successfully applied in coordinated downscaling experiments, including land use changes in the context of the CORDEX FPS LUCAS (Land Use and Climate Across Scales) (Davin et al., 2020; Breil et al., 2020; Sofiadis et al., 2022; Mooney et al., 2022; Daloz et al., 2022).

## 185 2.3 Structural changes in REMO2020

There are some changes to the overall structure (order) of the physics calculations in REMO2020 compared to previous versions of REMO. These changes include, for example, separating the cloud cover calculation from the stratiform cloud scheme, separating the albedo calculation from the radiation model, and shifting all of the above to be calculated earlier in the physics routine. These changed were also made to make the code structure more stable, efficient, clear, and consistent with the rest

of the physics calculations. In addition, the code has been made more modular to facilitate existing component updates and support easier implementation of new components. This supports any new technical requirement arising from climate service needs or the CORDEX initiative; for example, urban modelling (Langendijk et al., 2024).

In REMO2020, corrections have been implemented to address errors in the discretization of condensate fluxes within the convection scheme, as well as inconsistencies in the treatment of convective detrainment introduced by Mauritsen et al. (2019). Moreover, following the findings of Vergara-Temprado et al. (2020), who showed that switching off deep convection at spatial resolutions of  $\leq 25$  km can lead to an improved model performance, we revised the convection code such that different convection parameterizations (deep, mid-level, shallow, and cold) can be switched off separately. While the original subgrid convection parameterization of Tiedtke (1989) allows for different convection sub-parameterizations to be switched off individually, enabling these switches in our work required code modifications due to REMO-specific changes.







In addition to the structural changes to the stratiform cloud scheme, cloud cover calculation, and cloud droplet concentration, the affiliated codes were completely rewritten in REMO2020. REMO2020 introduces the latest ECHAM6 style coding structure and error fixes (Giorgetta et al., 2013; Stevens et al., 2013; Mauritsen et al., 2019), for both all cloud sub-modules, while keeping REMO-specific requirements. These needs are mainly related to different structures in physics calculations. Despite REMO2020 almost being completely rewritten, the new sub-modules are more of an update to the existing schemes, and the code-level changes were mainly targeted to faster performance of the code itself.

In REMO2020, the approach to radiation has changed slightly. Previously, short- and long-wave (SW/LW) fluxes at the surface were partially based on information that was updated on each radiation time step, usually once per simulated hour, and often used the grid-mean values of different tiles. For example, the SW radiation budget at the surface was based on grid-mean albedo values for all tiles. Although this approach is valid, it can lead to errors when a tile — such as a lake — needs to compute its own shortwave (SW) radiation budget. In the new approach, each tile calculates the SW budget separately and the averaging is done afterwards for grid-mean variables. Comparably, the LW budget had similar issues; the outgoing surface LW flux was calculated on each time step, but it was based on grid-mean surface temperature. The new model version allows the tile-wise LW budget calculation. This is very important, for example, when a grid box has open sea and frozen lake fractions. In such cases, the tile-mean outgoing LW flux can cause unrealistically high cooling over lakes, as the dominant flux from the open sea skews the mean values. In addition, this approach reduces artificial LW cooling between the radiation calls because the surface outgoing LW flux for tiles is updated based on the surface temperature at each time step. This improves the tile-wise LW budget and reduces artificial errors. Overall, the new SW and LW approach allows the model to be more responsive to changes in tile/fraction variables that directly influence the surface radiation budget and increases stability in specific cases, as mentioned above.

The overall structure for tiles has also been updated. The model still uses the three default tiles (land, sea, and frozen sea), but adding new tiles has been simplified. The code-level calculations are now more straightforward and fully automatic for different tiles. The different tiles are directly linked to their specific calculation methods by using procedure pointers; for example, the roughness length, latent and sensible heat, surface humidity, and virtual temperature calculations are done through a common interface, which actually points to the specific separate tile calculations. This also means that the lake model FLake

implementation in REMO2020 had to be rewritten (Pietikäinen et al., 2018). FLake is now a so-called passive module, which means that the module itself and the related components needed by the FLake module will be compiled and linked only if FLake is activated in the configuration phase. If FLake is set to be active, it automatically adds the lake tile to the default list of tiles and builds the necessary predefined interfaces with the main model.

Similarly to FLake, the iMOVE sub-model in Section 2.2, was implemented as a passive module to REMO2020. In addition, we have revised the coupling structure and improved the overall iMOVE performance in terms of computational efficiency. The iMOVE configuration is currently used in the WCRP CORDEX FPS LUCAS initiative (Rechid et al., 2017) and served as the starting point for adding an irrigation parameterization, as a new tile, to REMO (Asmus et al., 2023).

In addition to the these updates to the main branch of REMO2020, additional developments have created in separate branches including the inland glacier module by Kotlarski (2007), the online chemistry module by Teichmann (2010) and the online aerosol module by Pietikäinen et al. (2012, 2014). These branches are not yet included in REMO2020.

REMO2020 also includes an updated non-hydrostatic core following Göttel (2009). The technical details of its restructuring and a new Python programming language driven configuration, including active and passive modules, will be presented in future publications. In this paper, we focus on the updates to the physics modules, using the hydrostatic dynamical core, and their climate impacts.

In this work, special focus has been given to the tuning parameters of the model. The tuning parameters of REMO have not been updated before to such a large extent as it has been done in frame of this work. These parameters are used within various physical modules to adjust the main model results to better match some targeted features of the climate system (Mauritsen et al., 2012). We will not, however, go into great detail regarding the process of tuning, nor the different processes affected. Simply put, we checked existing tuning parameters of the model and made some adjustments for current usable resolutions, including both horizontal and vertical components. The results of both short and longer-term simulation were compared against measured climate and the best-matched combinations of the parameters were chosen. The modified parameters were all related to cloud and snow processes. For example, we changed the rate at which condensate is converted to precipitation in the convective updrafts, the entrainment rates of shallow, mid-level and deep convection, the fraction of the relative cloud mass-flux at the level above the non-buoyancy level, and the parameters controlling auto-conversion and accretion for large-scale clouds. For snow, we introduce the following tuning variables described in the next section.

#### 2.4 Snow modelling improvements







In simulations over the European domain, REMO has a tendency to have a cold bias in Northern Europe during the northern hemisphere winter (Kotlarski et al., 2014). The extent of the bias was partly hidden by the heat coming from the simple lake treatment, as shown by Pietikäinen et al. (2018). Possible reasons for the cold bias point to the snow physics, especially the snow heat conductivity. In this work, we changed the existing snow module to include 3 layers, and improved the snow heat conductivity and density to include more detailed parameterizations. Moreover, the snow radiation properties are improved, and the fractional snow cover calculation approach has been revised.

In this work, computational efficiency was a priority when improving the snow calculation. We chose to remove the artificial  $10 \, \text{cm}$  top approach explained in Sec.  $2.2 \, \text{and}$  instead implement a more physical multi-level approach for the snow pack. We also improved the density and heat conductivity approaches. The new snow module consists of  $3 \, \text{separate layers}$ . The top two layers have fixed heights and the lowest one can grow freely. The top layer has a maximum height of  $0.025 \, \text{m.w.e.}$  (meter water equivalent), corresponding roughly to  $10 \, \text{cm}$  of real snow, depending on the density. The second layer from the top has a maximum height of  $0.0525 \, \text{m.w.e.}$ . The top  $2 \, \text{layer}$  heights were optimized based on tuning simulations done for the snow module. Similar to the original approach, the residual surface energy fluxes only influence the top layer. The heat exchange between different snow layers and between the lowest snow layer and top soil layer is calculated with the same heat conductivity approach as used in the default 5-soil layer scheme (e.g., Semmler, 2002; Kotlarski, 2007). When there is enough snow, the heat solver performs calculation over 8-layers. The solver needs information about layer properties such as density, heat conductivity, and layer height. This presents a small issue for the solver because in the original snow scheme of REMO both the snow density and snow heat conductivity depend on snow temperature  $T_{\rm sn}$  and increase with increasing temperature (Roeckner et al., 2003). Thus, the characteristics of snow heat exchange in the solver only depend on the snow temperature, omitting any other influencing factors.




To improve the heat conductivity solver in terms of snow layer heat exchange properties, we have changed the snow density and heat conductivity approaches. The 3-layer scheme works by first calculating the falling snow density  $\rho_{snfr}$  based on Vionnet et al. (2012); Lafaysse et al. (2017), which takes into account the meteorological conditions on falling snow:

$$\rho_{\text{snfr}} = \max(\rho_{\text{snmin}}, a_{\rho} + b_{\rho}(T_{\text{2m}} - T_{\text{melt}}) + c_{\rho}\sqrt{W_{10m}}),$$
 (2)

where  $\rho_{\rm snmin}$  is 50 kg m<sup>-3</sup>,  $T_{\rm 2m}$  is the 2-meter temperature in K,  $T_{\rm melt}$  is the melting temperature of water in K,  $W_{10m}$  is the 10-meter wind speed in m s<sup>-1</sup>,  $a_{\rho}$  is 109 kg m<sup>-3</sup>,  $b_{\rho}$  is 6 kg m<sup>-3</sup> K<sup>-1</sup>, and  $c_{\rho}$  is 26 kg m<sup>-7/2</sup> s<sup>1/2</sup>. Fresh snow falls into the top layer, and if it already has snow, the existing snow density  $\rho_{\rm sn}$  is first updated using a modified aging approach by Verseghy (1991):

$$\rho_{\rm sn}(t+1) = \rho_{\rm snmax} + (\rho_{\rm sn}(t) - \rho_{\rm snmax}) \exp\left(\frac{-0.01\Delta t}{3600 \, s}\right),$$
 (3)

where  $\Delta t$  is the time step in seconds and  $\rho_{\rm snmax}$  the snow maximum density in  $~\rm kg~m^{-3}$ . Based on work done by Brown et al. (2006), the snow aging in Eq. 3 is too fast for the early snow season and not fast enough for melting seasons. Brown et al. (2006) proposed a fix, that limits the  $\rho_{\rm snmax}$ . We have implemented that fix with some modified values (based on tests not shown here):

$$\rho_{\text{snmax}} = \begin{cases} \rho_a - (d_a/d_s) \cdot (1.0 - \exp(-d_s/d_b)), T_{\text{snow}} < T_{\text{melt}_{25}} \\ \rho_b - (d_a/d_s) \cdot (1.0 - \exp(-d_s/d_b)), T_{\text{snow}} \ge T_{\text{melt}_{25}}, \end{cases}$$
(4)

where  $\rho_a$  is 475 kg m<sup>-3</sup>,  $d_a$  is 20470 m,  $d_s$  is the height of the snow layer in meters,  $d_b$  is 67.3 m,  $T_{\rm snow}$  is the snow layer temperature in K,  $T_{\rm melt}$ -0.25 K and  $\rho_b$  is 725 kg m<sup>-3</sup>.

After updating the old layer density from previous time step, the final top layer density is calculated using an arithmetic mean from the fresh snow and old snow densities, with the layer thicknesses acting as weights (weights are the amount of snow coming from the layer above and the existing layer thickness). If the top layer thickness exceeds the layer maximum limit, the extra amount of snow is moved to the layer below. If the layer below has already snow from the previous time step, the density of the old snow is updated (aging) and the arithmetic mean is used again to calculate the layer density. This is repeated until the lowest level is reached, which can grow without any height limitation. After all densities are calculated/updated, the snow heat conductivity is calculated. In the 3-layer snow module, the snow heat conductivity is based on snow density as presented in Calonne et al. (2011).

The snow heat conductivity parameterizations from Sturm et al. (1997), Riche and Schneebeli (2013), and Calonne et al. (2011) were tested; the latter of which proved to be the best choice for REMO2020 after multiple test simulations. The new approach for snow density and snow heat conductivity, allows for the temperature solver to utilize updated values with fresh snow and aged snow as input throughout the calculations, leading to more realistic snow characteristics and overall improvements in the energy budget. After the updated temperature values are calculated for each layer, we check if there has been any melting of snow. This follows the original approach by Roeckner et al. (1996) and is now calculated for each layer separately. The new 3-layer snow model is only used by REMO, but similar multi-layer snow approaches are being used also in other RCMs.

In terms of the snow albedo, a modified version of the work done in Pietikäinen et al. (2018) was implemented in REMO2020. REMO2020 now has two snow albedo schemes: 1) the original temperature-dependent (e.g., Kotlarski, 2007) and 2) the Biosphere-Atmosphere Transfer Scheme (BATS; Dickinson et al., 1993). In Pietikäinen et al. (2018), only the BATS scheme allows for the visible (VIS, 0.25 - 0.68 µm) and near-infrared (NIR, 0.68 - 4.00 µm) range albedos for snow to calculate separately. In this work, the bandwidth separation of snow albedo is now automatic for VIS and NIR and independent of the scheme chosen. This means that even with the original temperature-dependent scheme, we can calculate snow albedo for VIS and NIR separately based on updated snow albedo limits. The updated values are: the snow albedo values for fully forested areas vary for VIS from 0.35 to 0.2 (earlier 0.4 to 0.3) and for NIR from 0.3 to 0.15. For pure snow and glaciers the values for VIS are from 0.8 to 0.4 (no changes) and for NIR from 0.6 to 0.2. These changes were motivated by previous works done by Gao et al. (2014); Pietikäinen et al. (2018) and especially for forested areas satellite measurement studies done by Hovi et al. (2019); Jääskeläinen and Manninen (2021). The final snow albedo is calculated from the limits shown above and the forest fraction (Kotlarski, 2007). By setting a namelist variable, the snow temperature and BATS snow albedo schemes can be used separately or together by weighting the final albedo between the schemes. In this work, we have used equal weights for both schemes. This approach was chosen based on many test simulations (not shown) and it is the default approach in REMO2020.

In addition to the albedo changes shown above, the total albedo of snow-covered areas is based on fractional snow cover (FSC). In previous REMO versions, the FSC was calculated by dividing the snow water-equivalent (SWE) value by 0.015 m (max value was set to 1.0). In REMO2020, FSC is now calculated based on Napoly et al. (2020). It calculates the surface heat fluxes and the surface radiation balance while accounting for the impact of vegetation. Since REMO2020 uses a single-layer forest canopy approach that does not calculate the forest snow skin reservoir, we do not separate the forest fraction for albedo

calculations and use the original approach for snowy forest albedo. Moreover, the FSC is used to calculate the total emissivity of the ground. The original emissivity is combined with a new snow emissivity value of 0.97 based on the works of Chen et al. (2014) and Cole et al. (2023).

## 2.5 Aerosol climatology






Previously, the default configuration for aerosol in REMO was the Tanré aerosol climatology (Tanré et al., 1984). This climatology is fairly old, has a coarse resolution, and lacks temporal dependency (e.g., Zubler et al., 2011). The absence of time-varying aerosols, especially in terms of anthropogenic aerosols, can negatively impact future projections Boé et al. (2020). Applying the interactive aerosol module by Pietikäinen et al. (2012) is still computationally too heavy for long production runs, such as those done within the CORDEX project (Giorgi et al., 2009; Jacob et al., 2020). Therefore, we have updated the aerosol tropospheric aerosol forcing climatology of REMO using the simple plume (SP) implementation of the second version (v2) of the Max Planck Institute Aerosol Climatology MACv2-SP (Kinne et al., 2013; Fiedler et al., 2017; Stevens et al., 2017; Kinne, 2019). MACv2-SP provides the aerosol optical depth (AOD), single scattering albedo (SSA), and asymmetry parameter (ASY). The anthropogenic part of MACv2-SP is based on the plume model and is included in the model at the code level. It is called on every time step and provides the spatio-temporal distribution and wavelength dependency of the optical properties of anthropogenic aerosols (Stevens et al., 2017). MACv2-SP also includes an option for an empirical fit for aerosol–cloud–albedo effects (Twomey effect) by providing the change in the cloud droplet number concentration (Fiedler et al., 2017; Stevens et al., 2017). It should be noted that MACv2-SP can also be used for scenario simulations (Fiedler et al., 2019), which is an important factor for future projections (Boé et al., 2020) and supports the CORDEX aerosol forcing protocol for CMIP6 downscaling (Solmon and Mallet, 2021; Katragkou et al., 2024).

For the stratospheric (volcanic) forcing, a similar approach as in CMIP6 (details in Thomason et al. (2018)) has been implemented. We have used the latest version 4 datafiles of stratospheric forcing, which cover the time period from 1850-2018. The data is on a 5° latitudinal grid and includes 70 vertical levels reaching up to 40 km. The extinction coefficient (EXT), SSA, and ASY are provided on a monthly scale for shortwave and longwave radiation separately, including different bandwidth ranges. If the simulated year does not match the data range, for example future scenario simulations, we use a background stratospheric aerosol approach. These values are based on 1999 to 2001 values and have been monthly averaged for EXT, while for SSA and ASY a weighted average mean using EXT as weights has been used. The file for REMO was prepared separately to take into account the different bandwidths used in the model. At the code level, the data is remapped to REMO's rotated lon-lat grid and the vertical coordinates are remapped to REMO's vertical coordinates, the latter being done on each radiation time step. The transformation from EXT to AOD is done by summing up the extinction multiplied by level height for all data levels belonging to each of REMO's vertical level, while SSA and ASY are averaged using EXT/AOD values as weights over the data levels used in each of REMO's vertical level. AOD, SSA, and ASY are then used normally in the radiation code, together with the MACv2-SP climatology. Hereafter, the combination of the natural MACv2.0 part, anthropogenic MACv2-SP part, and the stratospheric aerosol part will be called jointly as MACv2-SP climatology. It should be noted that MACv2-SP is also used by other RCMs and by many GCMs.

**Figure 1.** Seasonal mean aerosol optical depth (AOD 550 nm) from the AATSR satellite and from aerosol climatologies: The new MACv2-SP aerosol climatology, MERRA-2 climatology, and the old Tanré aerosol climatology for the year 2005. All data is on their native grid.

In addition to updating the aerosol climatology, we have also made the aerosol treatment in terms of the radiation scheme more flexible. Climatologies are implemented in a modular way and the SW and LW radiation code sub-modules automatically use the selected climatology (Tanré is still available in REMO, although used only for testing purposes from now on). It is also now possible to use external sources of aerosol parameters with a small change to the source code. For example, we have introduced the MERRA-2 aerosol climatology (Gelaro et al., 2017) following the CORDEX aerosol forcing protocol (Solmon and Mallet, 2021; Katragkou et al., 2024). This important new feature supports the ongoing CMIP6 downscaling activities within the CORDEX project.



Figure 1 shows how the seasonal evolution of AOD improved using the MACv2-SP and MERRA-2 climatologies compared to Advanced Along-Track Scanning Radiometer (AATSR) satellite 2005 data (Copernicus Climate Change Service, Climate Data Store, 2019). Both the MACv2-SP and MERRA-2 climatologies outperform the Tanré climatology. Small differences in

AOD, between the observed AATRS and the MACv2-SP/MERRA-2 climatologies, but the overall features and values are very realistic. This also holds true for domains other than Europe (not shown).

## 2.6 Prognostic precipitation



Precipitation in REMO's stratiform cloud scheme is calculated at each time step. As a model's spatial resolution increases, the time step decreases, invalidating the assumption that mass fluxes of the vertical column can be calculated entirely from top to bottom within a single time step, as is the case in REMO2020's cloud micro-physical scheme (Roeckner et al., 1996, 2003). This includes the representation of evaporation, auto-conversion and freezing (Roeckner et al., 1996, 2003).

Consequently, the extent which precipitation can travel downward within one time step must be determined. Some precipitation may need to remain in the atmosphere to be included in the calculations for the next time step. To overcome this problem, we have introduced a statistical precipitation sedimentation scheme by Geleyn et al. (2008) and Bouteloup et al. (2011). The three probabilities for precipitation sedimentation in the new scheme are: 1) precipitation is already present in the layer at the beginning of the time step, 2) precipitation arrives from the layer above, and 3) precipitation is formed within the layer during the time step. This means some precipitation stays within a layer and is treated in the next time step, thus the new scheme acts as a memory for precipitation. In practice, this means the model has separate 3-D fields for rain and snow for each time step, which includes the amount of precipitation that did not fall into the grid box below or to the ground. Between the time steps, the model undergoes the dynamical step, i.e., the advection of mass and energy. We have included the 3-D precipitation fields in the advection part of the dynamical shift, as well as into horizontal diffusion. The vertical diffusion is considered to be insignificant compared to the precipitation velocities and it is not calculated for the prognostic precipitation.

The precipitation flux represents a whole grid box, whereas processes like evaporation of rain and sublimation of snow depend on the fractional area of a grid box. This fraction of precipitation in a grid box follows the approach used in the ECHAM model (Giorgetta et al., 2013, and references therein). In this approach, the fraction is based on the precipitation flux coming to a layer and on the newly formed precipitation in the layer. When using the prognostic precipitation, the amount of precipitation from the previous time step must also be considered. Thus, the new approach is a modified version of the one shown in Giorgetta et al. (2013) and defines the fraction of precipitation  $C_{\text{pr}}^{\text{tk}}$  in layer k as follows:

$$C_{\text{pr}}^{k} = \begin{cases} \max(\hat{C}_{\text{pr}}, \frac{C_{t-1}^{k} P r_{t-1}^{k} + C^{k} P r_{\Delta}^{k} + \hat{C}_{\text{pr}} P r^{k \cdot 1}}{P r_{t-1}^{k} + P r_{\Delta}^{k} + P r^{k \cdot 1}}), & P r_{t-1}^{k} + P r_{\Delta}^{k} + P r^{k \cdot 1} > Cqt_{\min} \\ 0, & P r_{t-1}^{k} + P r_{\Delta}^{k} + P r^{k \cdot 1} \le Cqt_{\min} \end{cases}$$

$$(5)$$

where  $C_{t-1}^{\mathbf{k}}$  is the cloud cover from the previous time step in layer  $\mathbf{k}$ ,  $Pr_{t-1}^{\mathbf{k}}$  is the precipitation flux from the previous time step,  $C^{\mathbf{k}}$  is the fractional cloud cover,  $Pr_{\Lambda}^{\mathbf{k}}$  is the newly formed precipitation flux,  $Pr^{\mathbf{k}-1}$  is the precipitation flux above,

 $Cqt_{\rm min} = 10^{-12} \ {\rm kg \ s^{-1} \ m^{-2}}$  and  $\hat{C}_{\rm pr}$  is defined as follows:

$$\hat{C}_{\mathrm{pr}} = \begin{cases} \begin{cases} C_{\mathrm{pr}}^{\mathrm{k-1}}, & Pr^{\mathrm{k-1}} > Pr_{t-1}^{\mathrm{k}} \\ C_{t-1}^{\mathrm{k}}, & Pr^{\mathrm{k-1}} \leq Pr_{t-1}^{\mathrm{k}} \end{cases}, & Pr_{t-1}^{\mathrm{k}} > Pr_{\Delta}^{\mathrm{k}} \\ C^{\mathrm{k}}, & Pr^{\mathrm{k-1}} > Pr_{\Delta}^{\mathrm{k}} \\ C^{\mathrm{k}}, & Pr^{\mathrm{k-1}} \leq Pr_{\Delta}^{\mathrm{k}} \end{cases}, & Pr_{t-1}^{\mathrm{k}} \leq Pr_{\Delta}^{\mathrm{k}} \end{cases}$$

$$(6)$$

The scheme is computationally efficient and fully integrated into the current updated stratiform cloud scheme. The precipitation velocities used in the scheme were also updated and are based on Roeckner et al. (2003). Moreover, to overcome an issue of having too much rain above the freezing level in (non-hydrostatic) high-resolution simulations, the freezing rain approach by Doms et al. (2021) was implemented into the prognostic precipitation scheme. This, however, was not used in the simulations within this work. In terms of the convection, the prognostic scheme cannot be directly applied to the convection scheme. The direct convective precipitation does not have a precipitation memory, but the convective transported moisture will be handled by the stratiform scheme.

# 2.7 Time filtering


The time integration of REMO utilizes the leap-frog scheme with the Robert-Asselin (RA) time filter (Asselin, 1972). It is known, however, that the RA filter can introduce some errors and dampen the solution amplitude in non-linear cases. To mitigate these effects, Williams (2009) proposed the Robert-Asselin-Williams (RAW) filter, which potentially improves the accuracy significantly. In RAW filter a second dimensionless filter parameter  $\alpha$  is introduced to stabilize the leapfrog time-stepping scheme even further and reduce the amplitude error. Details how the RAW filter and leap-frog scheme actually function can be found from Williams (2009).

In the new REMO2020 version, users can choose between the original RA filter and the new RAW filter. The filter parameter  $\alpha$  is set in a namelist controlling the simulation and can be easily changed. In this work, we have defined the default value of  $\alpha = 0.75$  for REMO2020, based on multiple test simulations (not shown).

#### 2.8 Dynamical core and wetcore

The hydrostatic dynamical core of REMO is based on DWD's former NWP model EM. It handles the transport of energy and mass both vertically and horizontally. In the current model version, horizontal advection for water species (humidity, cloud water, cloud ice and optionally prognostic rain and snow) is done with an explicit upstream method, while vertical advection is handled with an implicit approach. The dynamical core is computationally efficient, but not mass-conserving. To address this issue, a mass fixer is used. The main dynamical core of REMO2020 slightly differs from previous versions and has been re-written with optimizations in mind. The structure for the transport/advection of humidity, cloud water and ice has been improved and we have added the prognostic precipitation tracers rain and snow (see Sec. 2.6).

In the work by Pietikäinen et al. (2012), the authors introduced an interactive aerosol module to REMO. As advection of aerosol species plays an important role, the authors implemented a mass-conserving, positive definite, and computationally efficient finite difference, anti-diffusive advection scheme proposed by Smolarkiewicz (1984, 1983). The implementation was based on earlier work by Langmann (2000) and Teichmann (2010). In REMO2020, the advection scheme from the aerosol version was revised and implemented inside the current dynamical core in a modular way allowing one to choose between the original approach, a new wetcore approach, or a mixture of these (currently, only the horizontal wetcore advections and original vertical advection combination is supported). In the wetcore advection approach, all water species (humidity, cloud water and ice, and rain and snow fall if prognostic precipitation is activated) are transported using the wetcore advection routines. All other dynamical core calculations, such as numerical diffusion, remain unchanged. If the wetcore is used with the explicit vertical advection, the implicit vertical diffusion occurs after the advection; otherwise, it is done together with the implicit vertical advection. Moreover, special focus was given to optimizing the advection routines to enhance their run-time speed.

The wetcore approach also essentially removes the need for the mass fixer, but it is still activated. A downside of the wetcore approach is the increased computational burden, which must be considered when choosing between the original and wetcore approaches, as will be shown later when analyzing the results (Sections 4.2.3 and 4.2.4).

#### 435 3 Simulations and Data





This Section describes the setup of our simulations and the observational data we have used in our analysis. We have performed simulations with REMO2015 and REMO2020 using a configuration with 27 vertical levels, similar to previous CORDEX activities for comparison purposes, as well as a REMO2020 configuration with 49 vertical levels, similar to the latest and forth-coming CORDEX activities. In the following analysis, references to REMO imply all versions (REMO2015, REMO2020<sub>27</sub>, REMO2020<sub>49</sub>), while references to REMO2020 imply REMO2020<sub>27</sub> and REMO2020<sub>49</sub>. Table. 1 summarizes the main configuration differences. For the evaluation of these model versions, various datasets, described below, were used.

## 3.1 Simulation Setup

Several REMO simulations were conducted for the EURO-CORDEX domain with a 0.11° resolution (leading to a gridbox size of 12.5×12.5 km²) for the period from January 2000 to December 2010. The first year was removed as it is treated as spin-up for the atmosphere, leaving a 10-year period for analysis. A warm-start method for soil and lakes was applied in all simulations (more details can be found, e.g., in Gao et al. (2014) and Pietikäinen et al. (2018)). The lateral meteorological 6-hourly boundary forcing employed is either ERA-Interim data (Dee et al., 2011) or ERA5 data (Hersbach et al., 2020). For years 2000-2006, the updated ERA5.1 data was used. Several configurations of REMO2020 were tested, including those with either 27 vertical levels (REMO2020<sub>27</sub>) or with 49 vertical levels (REMO2020<sub>49</sub>) with the model top reaching 25 km and 30 km altitude, respectively. All 27-level simulations used ERA-Interim lateral boundary data, while all 49-levels simulations used ERA-5. In this way, we can directly compare the 27-levels simulation between REMO2015 and REMO2020. When

Table 1. Different REMO simulations and their main configuration

| Simulation name                | Configuration                                                              | Lateral boundary forcing |
|--------------------------------|----------------------------------------------------------------------------|--------------------------|
| REMO2015                       | old default, 27-levels                                                     | ERA-Interim              |
| REMO2020 <sub>27</sub>         | new default, 27-levels: FLake, 3-layer snow, MAC2-SP, RAW filter, re-tuned | ERA-Interim              |
| REMO2020 <sub>27</sub> Shallow | new default, 27-levels with only shallow convection                        | ERA-Interim              |
| REMO2020 <sub>49</sub>         | new default, 49-levels: FLake, prognostic precipitation,                   |                          |
|                                | 3-layer snow, MAC2-SP, RAW filter, re-tuned                                | ERA5                     |
| REMO2020 <sub>49</sub> MERRA-2 | new default, 49-levels with MERRA-2 aerosol climatology                    | ERA5                     |
| REMO2020 <sub>49</sub> iMOVE   | new default, 49-levels with interactive MOsaic based VEgetation            | ERA5                     |
| REMO2020 <sub>49</sub> Wetcore | new default, 49-levels with explicit horizontal advection                  | ERA5                     |
| REMO2020 <sub>49</sub> Shallow | new default, 49-levels with only shallow convection                        | ERA5                     |

comparing REMO2020<sub>27</sub> with REMO2020<sub>49</sub>, part of the differences may come from the different lateral forcing. We did not repeat any of the 27-levels simulations with ERA5, because this configuration will not be used anymore.

The REMO2020<sub>49</sub> iMOVE simulation employs the PFT distribution of the year 2015 from the LUCAS LUC dataset v1.1 (Hoffmann et al., 2022, 2023), based on the ESA-CCI LC-derived LANDMATE PFT dataset (Reinhart et al., 2022), interpolated to the model grid. Irrigation was not considered in the simulations. Therefore, the PFTs "crops" and "irrigated crops" are aggregated into the REMO-iMOVE PFT "C3 crops".

As a reference for the older version of the REMO model, we use the results from the REMO2015 simulation (Jacob et al., 2012). It used 27 vertical levels and simulated the entire ERA-Interim period, but in this work, only the years 2001-2010 are analyzed. REMO2015 used an older configuration, which did not include, for example, the FLake lake module, and used the old Tanré aerosol climatology.

All REMO simulations used a relaxation zone for the 8 outermost grid boxes. This zone is excluded from our analysis to prevent the lateral forcing from directly impacting the results.

#### 3.2 Observational Data



For the evaluation of meteorological variables in REMO2020, we use the E-OBS dataset v30.0e as our reference (Cornes et al., 2018). The E-OBS data was remapped from its 0.1° regular grid to the coarser model grid, after the daily data was used to derive monthly and seasonal averages over a multi-year period. E-OBS has gaps in different areas for the time-period of our analysis. We did not include grid boxes with less than 21 days of data in a month when calculating the monthly averages for the analysis. REMO2015 and all REMO2020 simulation results are masked on a monthly basis based on E-OBS data, meaning that we use the model data only for those grid boxes where E-OBS has data. This should be kept in mind, along with as any underlying observational uncertainties (Jacob et al., 2014; Prein and Gobiet, 2017); particularly over Turkey, where the number

of data points is far less than in other areas. From the monthly results the differences are calculated and seasonal statistics are derived.

For high temporal and spatial resolution precipitation data over Germany we have used the Radar-based Precipitation Climatology Version 2017.002 RADKLIM product (Winterrath et al., 2018a, b). RADKLIM provides hourly precipitation data on a 1x1 km<sup>2</sup> grid. In this work, RADKLIM data was remapped to the REMO model grid resolution.

To evaluate the new 3-layer snow module performance, the ESA CCI Snow "SnowCCI" (European Space Agency Climate Change Initiative, Snow) v2 the snow water equivalent (SWE) dataset provided by Luojus et al. (2022) is used. SnowCCI is a satellite measurement-based 0.1° dataset. The slightly older v1 version has been recently compared with ERA5 and ERA5-LAND (Hersbach et al., 2020; Muñoz Sabater et al., 2021) products by Kouki et al. (2023). It should be mentioned that mountainous areas (Alpine regions) are masked in the dataset. We use the SnowCCI product also to compare the fractional snow cover (FSC; Nagler et al., 2022). SnowCCI FSC is available on a 0.05° grid and all SnowCCI products are remapped to the REMO model grid. The biases are calculated on a multi-year seasonal scale.





For the albedo and total cloud cover comparison, the 3rd edition of CLARA-A3 of the CM SAF CLARA satellite product (Karlsson et al., 2023) has been selected following Pietikäinen et al. (2018). CLARA-A3 is available on a 0.25° grid, to which all analyzed REMO results have been remapped for albedo comparison. For total cloud cover, we use the coarsest grid from ERA5, which is roughly 0.28°, and both CLARA-A3 and REMO data have been remapped to this grid.

The modelled vertical profile of cloud cover is compared with the satellite-based cloud fraction for different height levels obtained from CALIPSO-GOCCP (v3.1.2) (Chepfer et al., 2010). CALIPSO-GOCCP provides data from June 2006 to the end of 2020 on 40 vertical levels reaching from near-surface up to 19 km height, with a spatial resolution of 2°. The vertical cloud cover data from the simulations were transformed from the model levels to the same CALIPSO-GOCCP vertical levels. Tests using the full CALIPSO-GOCCP period versus only the overlapping time period between REMO simulations and CALIPSO-GOCCP were performed. Results indicated the full CALIPSO-GOCCP period could be used for the analysis as it allowed for more observational data with small differences. The same applies for the years used in the CALIPSO ice water content (IWC) dataset (NASA/LARC/SD/ASDC; Winker et al., 2024). This data was used to estimate the vertical structure of IWC from different model versions. The CALIPSO IWC dataset has 172 vertical levels reaching up to 20 km height, with a spatial resolution of 2/2.5°. Both CALIPSO-GOCCP and CALIPSO IWC datasets are used to show zonal mean vertical distributions, thus, in terms of spatial resolution, no remapping is needed. The global datasets, however, are limited based on REMO's real latitude and longitude coordinates to match the same spatial domain used in REMO.

The modeled leaf-area index (LAI) is compared with measurements from the Copernicus Climate Change Service (C3S) Climate Data Store (CDS) (Copernicus Climate Change Service, Climate Data Store, 2018), conducted under the SPOT Vegetation mission (SPOT-VGT). We use the version V1.0.1 actual LAI values on a 1x1 km<sup>2</sup> grid and remapped them to the REMO grid.

#### 4 Evaluation of Meteorological variables

In the following, several aspects of the models performance will be analyzed. The focus will be on the most popular variables by users of climate data but also on variables related to the changes done in the model.

#### **4.1 2-m** Temperature






The near surface 2-meter temperature of REMO2020<sub>27</sub> shows overall better agreement with E-OBS data than REMO2015, as seen in Fig. 2. The central/south-eastern warm bias observed in REMO2015 has reduced in REMO2020<sub>27</sub> during spring and summer, remains similar in autumn, and is slightly increased in winter. The cold bias in the north during winter and spring in REMO2015 has reduced in REMO2020<sub>27</sub> for winter but increased for spring. It is important to note that REMO2015 simulations were based on our old approach for lake temperature and icing condition (details in Pietikäinen et al. (2018)). This means that REMO2015 has artificial warming from lakes during colder months, which masks the cold bias seen in Fig. 2. When we introduce a lake model and remove the artificial heating from the lakes, the cold bias increases, as shown in Pietikäinen et al. (2018). Therefore, the reduced northern cold bias in winter and spring in REMO2020 simulation represents a greater improvement than apparent in Fig. 2.

With REMO2020<sub>49</sub>, the autumn warm bias has slightly increased and spread to northern Europe, but it has vanished in summer and reduced in spring. In winter, it is similarly enhanced in REMO2020<sub>49</sub> as in REMO2020<sub>27</sub>, and has also slightly increased over Western Europe. Summertime temperatures in REMO2020<sub>49</sub> are slightly too low throughout most of the domain, but the bias is small and the temperatures are much closer to measurements than with the 27-level versions. Using the MERRA-2 aerosol climatology shows small differences compared to the default MACv2-SP, but it does show improvements in the autumn warm bias (reduced). The MERRA-2 simulation can be considered more realistic in terms of aerosols, and the small difference indicates that the MACv2-SP approach captures the main features for our simulated time period. Longer simulations will be conducted within the EURO-CORDEX project to analyze how well the impact of trends in aerosol concentrations is captured by REMO2020 using both aerosol climatologies.

With the interactive vegetation version iMOVE, the winter cold bias in the north has almost vanished and is reduced in spring. We will discuss more about the albedo changes in Section 4.4, but it can be said that iMOVE reduces the positive albedo bias (reduces reflectivity) over the northern domain, contributing to the reduced cold bias seen in Fig. 2. In contrast, the iMOVE version has a warm bias in autumn, which is the highest of all simulations. This was also reported in the earlier iMOVE version by Wilhelm et al. (2014). It is linked to crop harvesting, which leads to too low albedo in the model. We have tried to improve this by slowing down the LAI decrease at the end of the harvesting season (Sec. 2.3), but evidently more work is needed to reduce the warm bias. We will also discuss this issue later in Section 4.6. Other than these, the iMOVE version also shows slightly warmer temperatures in central Europe than other model versions but has very well captured summertime temperatures throughout the domain.

The 2-meter daily minimum and maximum temperatures (Appendix Figs. A1 and A2) provide more insights into the model biases seen in Fig. 2. Overall, the 2-meter minimum values are too high, except in Northern Europe during winter and spring.

**Figure 2.** Seasonal mean 2-m temperature from E-OBS dataset and biases from different model versions. The seasonally averaged results are for the time period of 2001-2010.

The 2-meter maximum temperatures are too low, with some exceptions in Central Europe. The improved summertime Central Europe bias in the 49-levels simulation is due to the reduced 2-meter minimum and maximum biases. These changes indicate changes in cloudiness, which will be discussed further in Sec. 4.5.1. The autumn Central European warm bias mainly comes from the too high 2-meter minimum temperature, although the maximum temperature is also too high. The bias in the latter is smaller with 49-level simulations, except with the iMOVE simulation.

The winter 2-meter minimum temperature bias is strongest in REMO2015, followed by REMO2020<sub>27</sub>, while the smallest bias can be found in both REMO2020<sub>49</sub> simulations. The same pattern is seen for the winter 2-meter maximum temperature bias, but the amplitude is much smaller. The better-modelled winter minimum temperature is the biggest contributor to the decreased cold bias in REMO2020 simulations, although the better representation of maximum temperatures also plays a role. Spring has similar features in minimum temperature as winter, but the amplitude of the bias is much smaller. The maximum 2-meter temperatures in spring behaves differently than in winter: REMO2020<sub>27</sub> and REMO2020<sub>49</sub> have the highest biases, whereas REMO2015 and REMO2020<sub>49</sub> iMOVE have the smallest, yet still being too cold. Although the snow scheme and snow albedo approach have improved in the new version, there is still room for better representation of snow, which can be one explaining factor for the cold bias in both seasons in the northern domain. Evidence of the impact of soil properties on the spring cold bias will be shown in Section 4.4. Additionally, REMO does not have a detailed forest canopy model, which will influence the temperatures over forested areas and could partly explain the north-eastern cold bias (Haesen et al., 2021, please also note the Corrigendum).

## 4.2 Precipitation






In the following sections, the precipitation characteristics are analyzed in details. We show monthly plots for Europe and Central Europe, connect precipitation changes to temperature changes, and analyze precipitation distributions.

# 4.2.1 European scale

The differences in precipitation between different model versions and E-OBS data are shown in Fig. 3 (the relative differences are shown in Fig. A3). REMO2015 has both dry (south-western) and wet (north-eastern) areas, and REMO2020<sub>27</sub> behaves similarly, although the biases are slightly reduced. REMO2020<sub>49</sub> versions show more realistic results, but have a more systematic tendency to be too wet over mountainous areas. Seasonally, the winter dry bias in REMO2015 near coastal areas is gone in REMO2020 versions, but they have some excess precipitation on the eastern coast of the Adriatic Sea. During spring, the situation is similar to winter, but the Adriatic Sea excess is smaller. In summer, the Central European dry bias in REMO2015 is almost gone in REMO2020<sub>27</sub> and completely gone with REMO2020<sub>49</sub>. Finally, the autumn time follows a similar pattern: Central European dry bias is reduced and almost gone in the REMO2020 versions, while the mountainous areas have a wet bias.

**Figure 3.** Like in Fig. 2, but for seasonal absolute precipitation differences. The same plots with relative differences can be found in the appendix (Fig. A3).

## 4.2.2 Changes with 2-meter temperature






We also show the 2-meter temperature and precipitation biases in different Prudence regions (Christensen and Christensen, 2007). Figure 4 illustrates that seasonally, REMO2015 and REMO2020<sub>27</sub> are overall better at representing the 2-meter temperature (x-axis) than REMO2020<sub>49</sub> with any configuration, with winter in the Scandinavian Prudence region being an exception. REMO2015 and REMO2020<sub>27</sub> have more separation (clearly different biases), especially in southern Europe, compared to the different REMO2020 49-level versions. Both 27-level versions give similar results in winter and autumn, while during spring and summer REMO2020<sub>49</sub> is in better accord with E-OBS data. Similarly, REMO2020<sub>49</sub> outperforms the 27-level versions in spring and summer, especially in Southern Europe, while there is more discrepancy over northern Europe. During winter and autumn, REMO2020 with 49 levels has a clear tendency to be too warm, with some exceptions (iMOVE version of being the warmest), which can also be seen in Fig. 2.

In terms of precipitation biases, Fig. 4 shows the same information as in Fig. A3, but also reveals some interesting points. REMO2020 with 49 levels clearly has a wet bias over mountainous regions, as discussed before, but has much fewer dry biases than REMO2015 and REMO2020 with 27 levels. This means that the mean values shown in Figs. A3 and 4 tend to favor the 27-level simulations, as the spatial means also take into account the dry grid boxes, which in some cases have quite high values, skewing the mean. The opposite is visible for Eastern Europe, where all model versions have similar biases and show very little differences in precipitation in Fig. 4. The mountainous regions, like the Scandinavian and the Alps domains, again show the highest biases between different vertical level versions in winter and autumn, while in spring and summer, they are less, even being smaller with 49 levels in the summertime Scandinavian domain. The differences are not that high, reaching maximum about 0.8 mm/day in SON over the Alps, otherwise staying under 0.4 mm/day. And this even after considering that the 27-level versions have more dry biases than the 49-level versions. The 49-level simulations perform really well for non-mountainous regions, but they do have an issue with the mountainous areas. The relative differences shown in Fig. A3 basically show the same information, but also reveal more about how patchy the precipitation pattern is with 27-levels and how the mountainous region excess with 49 levels is not relatively that much higher. Overall, Figs. 2, A3, and 4 show that REMO2020, especially with 49 levels, is better at capturing the measured temperatures and shows clear improvements in precipitation, except over mountainous regions, where it has a clear wet bias. It should be kept in mind that these areas are also challenging for precipitation measurements and errors do occur, especially in sparser measurement network areas, like mountains (Bandhauer et al., 2022). The need for undercatch correction in the underlying gauge measurements data of E-OBS can also lead to overestimation when compared to gridded data (Hagemann and Stacke, 2023).

#### 595 4.2.3 Central Europe and convection

Zooming into Central Europe allows us to see the impacts of convection parameterization configuration and how the different advection and precipitation approaches influence the results. Figs. 5 and 6 are based on the same data as Fig. 3, but with a zoom into Central Europe and shown separately for 27 and 49 levels. Fig. 5 shows how REMO2015 has orographic biases over Germany, southern France, the Alps, and Italy. These are very visible during winter, but exists also during other seasons.

**Figure 4.** Differences in 2-meter temperature (x-axis) and daily precipitation (y-axis) between different REMO versions and E-OBS data for different Prudence regions. The data covers the whole simulated 2001-2010 period.

Figure 5. Central European seasonal precipitation from E-OBS dataset and the biases of different REMO2020<sub>27</sub> versions.

REMO2020<sub>27</sub> has these same features, but the magnitude of the biases is significantly lower. The main reason for the improved performance is the updated transport of cloud water and ice (Section 2.8). REMO2015 and REMO2020<sub>27</sub> with full convection (Tiedtke, 1989) show chessboard-like features over mountainous areas, especially during summer. As mentioned before, Vergara-Temprado et al. (2020) showed that switching off deep convection can lead to better performance of different model skills related to precipitation. We tested this with REMO2020 and Fig. 5 shows that with 27-levels only using shallow convection indeed improves the results and removes the chessboard-like pattern. The biases are more localized, and the results looks more realistic. There are, however, factors influencing how realistic the precipitation actually is beyond what Fig. 5 reveals and these will be discussed later on.

Similarly to the 27-levels, REMO2020<sub>49</sub> improves the precipitation biases when we zoom into Central Europe, as seen from Fig. 6. The orographic biases seen over Germany, southern France, the Alps and Italy in REMO2015 and partly in REMO2020<sub>27</sub> are now completely vanished, as is the chessboard-like pattern over mountainous areas. Over mountainous areas,


however, REMO2020<sub>49</sub> shows clear excess precipitation in winter, spring and autumn. The bias is clearly visible but does not stand out as a major issue in relative differences (Fig. A3). By default, for 49 levels, we use the prognostic precipitation scheme (Sec. 2.6), although the results seen in Fig. 6 would indicate the contrary, as the mountainous excess is almost entirely gone. The reason why we still use the prognostic scheme for 49 levels will be explained later. When we use only shallow convection for REMO2020<sub>49</sub>, the model becomes extensively too wet, behaving differently than with 27 levels. Fig. 6 also shows results from the explicit wetcore approach and it does not differ much from the default configuration in REMO2020<sub>49</sub>. The minor differences between the default REMO2020<sub>49</sub> configuration and the wetcore approach mean that the default configuration can be used without the much more computationally expensive wetcore, at least on hydrostatic-scale resolutions. It also means that the re-structured dynamical core performs very well for water species, even when compared to the wetcore approach.

#### 4.2.4 Precipitation probability distribution






Based on Figs. 5 and 6, we should consider using only shallow convection for 27 level and not activating the prognostic precipitation for 49 levels. This is, however, not the full picture, as can be seen in Fig. 7. It shows precipitation distribution for the whole modelled period from different model configurations over Germany and measured RADKLIM data on its native and REMO grids. When using the coarser resolution, the tail of the higher resolution (native) distribution naturally vanishes. Since all the model results are on the coarser REMO grid, the native RADKLIM values can be considered as maximum values for the coarser grid. Fig. 7, however, shows that most of the model versions have a higher number of high-intensity events than the native grid in RADKLIM. The worst two configurations are REMO<sub>27</sub> using only shallow convection and REMO2020<sub>49</sub> without the prognostic scheme, which showed more promising results earlier. Noteworthy is that we use a limit of 100 mm/h in Fig. 7 for the x-axis, and in reality, REMO<sub>27</sub> using only shallow convection and REMO2020<sub>49</sub> without the prognostic scheme have even higher extreme precipitation events than shown. As can be seen, the frequency of such events is not high, which explains why we don't see their influence on seasonal biases (Figs. 3, 5, and 6).

Figure 7 also shows that the older model version REMO2015 already had a tendency for too intense precipitation events. Almost the same can be said about REMO2020<sub>27</sub>. These two somewhat follow the high-resolution RADKLIM distribution, which should not be the case for coarser resolution simulations (e.g., Lind et al., 2016), but still makes their results more realistic than REMO<sub>27</sub> using only shallow convection and REMO2020<sub>49</sub> without the prognostic scheme. REMO2020<sub>49</sub> and REMO2020<sub>49</sub> with wetcore follow the RADKLIM coarse data very realistically and show no overestimation in extreme precipitation (realistically not even reaching the highest values). This again shows that our re-structured dynamical core performs very well and does not show any issues when compated to the wetcore approach. Moreover, even though the use of the prognostic precipitation scheme show excess precipitation over mountainous regions (Fig. A3), it gives more realistic results in terms of precipitation distribution.

Fig. 7 suggests that for REMO, we already have reached a resolution where - besides the sub-grid convective parameterization - the model starts to partly resolve convection, entering the so called grey-zone. This starts at lower resolutions than previously considered, supporting the findings by Vergara-Temprado et al. (2020). REMO2015 and REMO2020<sub>27</sub> still produce good enough results, although the extreme distribution tail is skewed towards unrealistically high extremes. REMO2015

Figure 6. Central European seasonal precipitation from E-OBS dataset and the different 49-levels model version biases against it.

with older tuning does not differ that much from REMO2020<sub>27</sub>, because they were both used with 27-level vertical resolution, which limits the impact of the convective parameterization. When we then switch to 49-levels, the tuning of the convective cloud scheme becomes more important. Furthermore, with 49-levels we move deeper into the grey-zone, i.e., REMO starts resolving parts of convection. The original approach, where the convective parameterization first does the mass-flux calculations and then the stratiform cloud scheme reacts to the state of the atmosphere within one time step starts to become invalid.

This together with better resolved vertical motion causes the model to have very extreme precipitation (Fig. 7, REMO2020<sub>49</sub>)

**Figure 7.** Distribution of hourly JJA precipitation sums over Germany from RADKLIM product (on 1x1 km<sup>2</sup> and EUR-11 grids) and from different model versions.

no-Prog) while the multi-year seasonal patterns look reasonable (Figs. 3 and 6). REMO2020<sub>49</sub> with only shallow convection does not improve the situation in Fig. 7, which is also the case in Fig. 6. When using 0.11° spatial resolution with 27 vertical levels, the vertical resolution was the limiting factor for the model not being inside the convective grey-zone, but increasing the vertical resolution to 49 levels pushed it there. In our current setup, this issue is solved with the prognostic precipitation (precipitation memory) and better tuning. Although the distribution in Fig. 7 looks much more realistic for REMO2020<sub>49</sub>, it still suffers from an excess of precipitation over mountainous regions. It should be mentioned that these regions are also challenging for gridded datasets, but clearly we do have too much precipitation, even considering this.




Naturally, the question arises of how well the conclusions from Fig. 7 hold in other areas than Germany. We do not utilize hourly measurement data for other regions but plotted the precipitation distribution from different model versions for the Prudence regions in Fig. 8. A very similar message can be seen from Fig. 8 as from Fig. 7; REMO2020<sub>49</sub> has the lowest extremes with the wetcore approach, and without prognostic precipitation, we get unrealistically high extreme values. The same can be said about REMO2020<sub>27</sub> with shallow convection only, whereas REMO2020<sub>27</sub> default configuration shows very similar results to REMO2020<sub>49</sub> in the British Isles, the Alps, Iberian Peninsula, and Mediterranean. REMO2015 gives higher extremes than REMO2020<sub>27</sub> and in some cases very high values, pointing to better performance of the new version, even with 27 levels. It should be mentioned that to get the more realistic precipitation distributions shown in Figs. 7 and 8, the convective cloud parameterization was tuned in terms of the detrainment rates. We took into account the spatial and vertical resolution changes together and decreased the detrainment rates accordingly, i.e., assumed that the model's step deeper into the grey-zone meant that we already have a somewhat better representation of the air flows and could reduce the tuning parameter values controlling them.

Figs. 5 and 6 also tell us something about the wet bias in 49-level simulations over mountainous regions. If we first concentrate to the 27-level simulations and look at the results over the Alps (also from Fig. 4), we see that the new model version is slightly wetter in winter but shows more realistic or similar results in other seasons. The dry biases in REMO2020<sub>27</sub> are

Figure 8. Distribution of hourly JJA precipitation sums over different Prudence regions from different model versions.

smaller, but the differences in areal bias over the Alps do not show this, meaning we did not only shift the model to precipitate more, but also improved the precipitation itself. The 49-level simulations have a clear wet bias over the Alps (and other mountainous regions), but the difference comes from much smaller areas. With 27 levels, especially during summer, the chessboard-like pattern is spread over a vast area in the Alps and has both very wet and very dry grid boxes. It is very obvious that convection plays a big role in the 49-level simulation biases. If we switch off the prognostic scheme, i.e., cloud water memory, we get really nice spatial patterns (Fig. 6), but the precipitation extremes get unrealistically high (Figs. 7 and 8). As mentioned, the problems with convection only get worse with 49 vertical levels, and our model starts to overshoot the total precipitation amount in mountainous regions. The precipitation biases over non-mountainous areas are very realistic with REMO2020<sub>49</sub> and there is a real need for grey-zone convection parameterization for the mountainous regions. A similar prognostic approach to that used for stratiform clouds may also be necessary for the convective part. With even higher resolutions using the non-hydrostatic setup, this issue is removed as the convective parameterization is not used (convection permitting simulations). Higher resolution has been shown to improve many precipitation metrics over mountainous regions, for example, over the Alps (Pichelli et al., 2021). Similar results can be seen in non-hydrostatic simulations of REMO2020 (not shown), confirming the need for better resolved climate simulations to overcome the difficulties in the grey-zone.

#### 4.3 Mean sea-level pressure




The mean sea-level pressure (MSLP) biases against E-OBS data from different model versions are shown in Fig. 9. REMO2015 and REMO2020<sub>27</sub> shows lower MSLP in all seasons and overall similar features. It is important to remember that both ver-

sions used the older ERA-Interim data as lateral forcing, whereas all REMO2020<sub>49</sub> versions used ERA5 forcing. REMO2020<sub>49</sub> shows better agreement with E-OBS in all seasons, and the low-pressure bias has reduced significantly. For winter, REMO2020<sub>49</sub> shows the least improvements and the low-pressure bias is visible, but with the iMOVE version, it is already very small. However, iMOVE shows the strongest low-pressure bias of the 49-level versions in summer. All REMO2020<sub>49</sub> versions also show some high-pressure bias for the Nordic countries in spring and some during summer for Sweden and Norway. In summary, REMO2020<sub>49</sub> shows good agreement with the measurements, and the new version outperforms the older version, even with the same 27-level configuration and the same driving data (ERA-Interim).

#### 4.4 Snow cover



Figure 10 presents the multi-year mean SWE biases from REMO2015, REMO2020<sub>27</sub> and REMO2020<sub>49</sub> against the SnowCCI data for January, February, March, and April. Here, the masking of Alpine regions in SnowCCI SWE data shows as bias-less areas in Western and Northern Norway and this limitation should be kept in mind. REMO2020<sub>49</sub> performs better in almost all regions, though there is some underestimation of SWE in northern Finland and Sweden, particularly during March and April. REMO2020<sub>27</sub> exhibits smaller biases in these areas but tends to overestimates SWE in the northeast, similarly to REMO2015. The excess precipitation in mountainous regions during winter (Fig. A3) leads to a slight but noticeable overestimation of SWE in these areas across all REMO2020<sub>49</sub> simulations. This overestimation is least pronounced in the iMOVE version, although the difference to REMO2020<sub>49</sub> is small. Overall, Fig. 10 indicates that the new 3-layer snow module does not reproduce unrealistic values; on the contrary, the new version outperforms the old one in terms of SWE, especially with the 49-level versions. While we do not show the actual snow height here, it is worth noting that the new version calculates it from three layers, each with its own prognostic density approach. This also enhances the heat exchange calculations, which are now performed separately for each layer, resulting in a reduced cold bias in the northern part of the domain (Fig. 2).

Daloz et al. (2022) evaluated the WCRP CORDEX FPS LUCAS models concerning the snow-albedo effect. An earlier version of REMO-iMOVE was one of the participating models and demonstrated very realistic snow cover results compared to MODIS-AQUA satellite observations. We have also examined the snow cover from the new model versions (essentially fractional snow cover extent), and the results are very realistic. Instead of reiterating those results, we focus here on the new variable fractional snow cover (FSC), which provides better insight into the snow-vegetation partition within the model. We cannot show results from REMO2015, as it did not calculate the FSC, but rather the fractional snow cover extent.

FSC is generally well captured by different model versions, except for the north-eastern part of the domain, as shown in Fig. 11 (note the one-month shift compared to Fig. 10 due to data coverage limitations). There is some underestimation for Finland, Sweden, southern Norway, and European Russia, especially in late winter, but the most significant issue is the overestimation over European Russia. REMO2020<sub>49</sub> with iMOVE, however, shows very little overestimation and provides more realistic results. It is important to remember that the FSC is inversely proportional to the surface roughness length (see Sec. (2.4). In REMO2020 and REMO2020<sub>49</sub>, the surface roughness length calculations for vegetation are based on monthly varying land surface parameters (Rechid and Jacob, 2006), whereas in iMOVE, they are based on interactively changing vegetation. In the latter, the surface roughness length is higher during winter for European Russia, leading to smaller FSC and better

**Figure 9.** Seasonal mean sea-level pressure from E-OBS dataset and the different model version biases against it. The seasonally averaged results are for the time period of 2001-2010.

Figure 10. Multi-year monthly SWE from the SnowCCI data (Luojus et al., 2022) and different model version biases against it.

representation compared to the SnowCCI satellite product. The more realistic snow cover distribution reduces the cold bias in these regions in MAM due to the FSC's impact on the surface heat fluxes and the surface radiation balance. It should be noted that the underestimation of FSC in the northern parts and overestimation over European Russia directly relate to the SWE biases shown in Fig. 10. However, the mountainous areas are an exception, as they tend to show more realistic results, despite having too much snow on the ground.

Figure 11. Multi-year monthly FSC from the SnowCCI data and different model version biases against it.



Figure 12 illustrates how SWE and FSC influence the surface albedo. We observe underestimation in the northernmost land areas and overestimation over European Russia, reflecting the biases from SWE and FSC. When the modelled SWE and FSC values are too low, the modelled surface albedo is also too low (northern areas). Besides the direct snow influence, the current single-layer forest canopy approach, which does not calculate the forest snow skin reservoir, impacts on the winter-time forest albedo. This is not, however, very visible when examining the albedo biases. Over European Russia, the overestimation in modelled SWE and FSC results in a surface that is too bright. This is not the case with iMOVE, where SWE and FSC are more realistic, and the albedo values are similarly in better agreement with the CM SAF CLARA-A3 satellite product. The same applies to Eastern Europe, where the model tends to have too low albedo values; using iMOVE, the bias is much smaller than with any other REMO configuration. As described in Sect. 2.2, unlike the standard REMO2020 land surface scheme, where albedo values are prescribed, iMOVE computes the albedo dynamically. For calculating the albedo value, iMOVE combines the dry soil albedo, based on the soil distribution from the Harmonized World Soil Database (HWSD), and the albedo values from

Figure 12. Multi-year monthly albedo from the CM SAF CLARA-A3 data (Karlsson et al., 2023) and different model version biases against it.

MODIS (Tsvetsinskaya et al., 2002) with a soil moisture dependency (Wilhelm et al., 2014). Furthermore, iMOVE accounts for litter albedo by incorporating the dynamically evolved and PFT-specific LAI values (Wilhelm et al., 2014).

The 2-meter temperature biases in Fig. 2 realistically follow the albedo features in Fig. 12. The northernmost parts, however, exhibit a cold bias despite the low albedo. This can be partly explained by the low solar radiation intensity linked to short daytime, thus limiting the albedo influence, and possibly also by the previously discussed missing influence of the forest

canopy heating effect. Additionally, lake surfaces tend to be too bright with REMO2020 in late spring. This feature was also reported by Pietikäinen et al. (2018) and is linked to the winter and springtime cold bias in the model, which delays the melting of the ice surface in the lakes. In REMO2015, this is less visible because it uses the closest sea point icing conditions, which bring other issues, such as artificial heating, as discussed in Sect. 4.1 and also reported by Pietikäinen et al. (2018).

#### 4.5 Clouds

760

765

In the following sections, cloud cover will be analyzed both spatially and vertically. The vertical profiles of cloud water and ice content will also be evaluated.

## 4.5.1 Cloud Cover

Figure 13 presents the multi-year monthly total cloud cover (TCC) from CM SAF CLARA-A3 data (Karlsson et al., 2023), along with the differences compared to ERA5 re-analysis, REMO2015, REMO2020<sub>27</sub>, and REMO2020<sub>49</sub>. ERA5 consistently underestimates the TCC over oceans and seas, regardless of the season. All versions of REMO exhibits the same tendency, with biases significantly larger than those found in ERA5. Over the Mediterranean Sea, both ERA5 and REMO underestimate the TCC in all seasons except summer. During summer, model biases are generally minimal, except for both REMO versions with 27 vertical levels, which exhibit notable overestimations — particularly over the eastern Mediterranean region. Over land, the different model versions exhibit a more complicated pattern. In REMO2015, the TCC over Continental Europe is underestimated in all seasons compared to CM SAF, with the greatest underestimation east of the Adriatic sea. Over Northern Finland and Northern European Russia, REMO2015 overestimates TCC in all seasons except summer. These biases are similar to those in ERA5, which also overestimates the TCC in summer over Northern and Eastern Europe. In REMO2020<sub>27</sub>, the overall bias pattern is similar to REMO2015, with some cases showing stronger biases (e.g., wintertime Mediterranean negative bias) and other showing weaker biases (e.g., summertime Mediterranean positive bias). In REMO2020<sub>49</sub>, the TCC bias is reduced over Continental Europe in spring and summer, although the bias over Finland and Eastern Europe is exacerbated, similar to ERA5. Overall, the mean cloud cover of REMO2020<sub>49</sub>, which performs best in summer, remains underestimated compared to CM SAF Cloud Cover due to biases over the Atlantic Ocean, and the RMSE remains unchanged from REMO2015.

## 4.5.2 Cloud fraction

To better understand the cloud cover biases, the vertical distribution of clouds is evaluated. In Figure 14, the zonal mean vertical distribution of cloud fraction for CALIPSO-GOCCP (Chepfer et al., 2010) satellite data, ERA5 reanalysis, and the REMO model versions are compared.

In CALIPSO-GOCCP, the maximum cloud fraction is found in the lowest 3 km, hereafter referred to as low-level clouds, with the greatest amount between 40° N and 70° N. Above 5 km, CALIPSO-GOCCP shows a maximum cloud fraction peak between 35° N and 45° N and around 65° N in winter and spring, and between 45° N and 55° N in summer and autumn. In

**Figure 13.** Multi-year monthly total cloud cover from CM SAF CLARA-A3 satellite data and biases against it from ERA5 reanalysis and different model versions.

summer, the drying branch of the Hadley cell is reflected in the reduced cloud fraction throughout the atmospheric column near  $30^{\circ}$  N.

In ERA5, high-level cloud fraction is consistently overestimated compared to CALIPSO-GOCCP, except at the most southern latitudes, while the low-level cloud fraction is underestimated across all seasons and all latitudes. The absence of clouds at heights below 5 km south of 45° N is particularly noticeable in ERA5. All versions of REMO show this same pattern,

but during summer and autumn, the cloud fraction above 5 km is in better agreement with CALIPSO-GOCCP than ERA5. Considering the spatial distribution of cloud cover bias in Fig. 13, the underestimated clouds are over the Atlantic Ocean, Mediterranean, and Black Sea in both ERA5 and REMO, with REMO also showing fewer clouds over land areas north of the Mediterranean. Moreover, the low-level cloud fraction in all REMO versions, regardless of season, is underestimated compared to CALIPSO-GOCCP. The high-level cloud fraction in all REMO versions is similar to or slightly overestimated compared to CALIPSO-GOCCP, particularly at northern latitudes. Differences between REMO2020 and REMO2015 mainly occur at altitudes above 5 km, north of 35° N, where both REMO2020 configurations capture the gradient of cloud fraction better than REMO2015 when compared to CALIPSO-GOCCP. Below 5 km, REMO2015 shows more clouds, especially during winter and spring, which is in better agreement with CALIPSO-GOCCP and explains the excess in TCC in Fig. 13.

**Figure 14.** Seasonal zonal mean vertical distribution of cloud fraction for CALIPSO-GOCCP (Chepfer et al., 2010) satellite data, ERA5 and different model versions. The analysed period for CALIPSO-GOCCP is from 2006 to 2020 and for ERA5 and REMO from 2001 to 2010.

## 4.5.3 Cloud Liquid and Ice Water Content


We use the cloud liquid water content (LWC) in Figure 15 to investigate the lack of clouds throughout the atmospheric column in REMO south of 45° N, particularly in summer. The cloud liquid water content in ERA5 is consistent for a given altitude with the greatest amount between approximately 1-2 km. In winter, the highest LWC values in ERA5 are concentrated in latitudes south of 65° N. In summer, the drying branch of the Hadley cell is prominent south of 45° N and LWC exceeding 0.01 g kg<sup>-1</sup> is concentrated mainly north of 45° N up to an altitude of 3 km. It should be noted that ERA5 underestimates the low-level clouds compared to CALIPSO-GOCCP, implying that the ERA5 cloud liquid water content should be greater or that a greater cloud fraction should be diagnosed with this given cloud liquid water content.

Compared to ERA5, all REMO versions show LWC values reaching higher altitudes. The highest LWC values, usually between 45° N and 70° N and below 3 km, are larger in REMO than in ERA5. REMO2015 differs from REMO2020 by having less LWC at higher altitudes and the maximum values are closer in REMO2020<sub>49</sub> than in REMO2020<sub>27</sub>, with the latter having the largest LWC values of all. REMO2020 shows higher LWC values in summer and autumn south of 35° N at altitudes below 5 km, which can also be seen in the higher cloud fraction in Fig. 14. We observe that REMO, especially with REMO2020, has LWC south of 40° N between 2-6 km height, which translates into clouds in Fig. 14. Despite REMO2015 and REMO2020<sub>27</sub> having more LWC near the surface, however, we do not get similar cloud fractions as with CALIPSO-GOCCP.

Figure 15. Seasonal zonal mean vertical distribution of cloud liquid water content from ERA5 reanalysis data and different model versions.

Figure 16 shows the ice water content (IWC). CALIPSO and ERA5 are in good accord. All REMO versions show similar overall features to CALIPSO, but the models overestimate the IWC, especially the REMO2020 versions. The overestimation is strongest in the northern part of the domain and there is more IWC at lower altitudes in all model versions, including ERA, than in CALIPSO. South of 45° N, particularly in summer, IWC values are fairly similar between all REMO versions. The differences in cloud fraction in this area are clearly coming more from the LWC than IWC. Interestingly, REMO2020 versions show more ice than REMO2015, although during the tuning process, the threshold controlling the separation of cloud water and ice was changed so that REMO2020 should produce less ice. We did, however, make many other changes to clouds, as explained in Sec. 2.3 and 2.6, leading to many other impacting factors ultimately resulting in increased IWC.




**Figure 16.** Seasonal zonal mean vertical distribution of cloud ice water content from CALIPSO satellite data, ERA5 reanalysis data and different model versions.

It should be noted that in these simulations, the sea surface temperature (SST) is prescribed and taken from the driving data (ERA-Interim/ERA-5). Studies with ocean-coupled REMO have shown how the coupled model reduces the SST over the Mediterranean area in summertime, leading to reductions in precipitation (Parras-Berrocal et al., 2020; Cabos et al., 2020). Applying this knowledge to our results, it is possible that the missing atmosphere-ocean coupling and its influence on SST leads to too high precipitation, less low-level clouds, and a biased TCC with an erroneous vertical profile. The Mediterranean Sea is located exactly where all REMO versions have most problems with missing low-level clouds and a coupled ocean-model approach has the potential to be a part of the solution.

Figure 17. Multi-year monthly leaf-area index biases from REMO2020<sub>49</sub> against satellite-based product SPOT-VGT.

# 4.6 Vegetation


Previous sections have demonstrated some of the benefits of using the interactive vegetation module in REMO. Fig. 17 shows the LAI from satellite data and the biases from different model versions. REMO2015, REMO2020<sub>27</sub>, and REMO2020<sub>49</sub> use the same static monthly-varying underlying vegetation map, and the differences in LAI results between the model versions are insignificant. Therefore, we only show the results from REMO2020<sub>49</sub>, which is our default configuration. Overall, REMO2020<sub>49</sub> overestimates the LAI in all regions except for Western Europe throughout all seasons. The reasons behind these differences stem from the input data (see Sec. 2.2) and the absence of a vegetation model. With iMOVE, the input data is updated to a very recent land cover dataset (Hoffmann et al., 2023), and the vegetation changes are interactively modelled. This improves the LAI biases of the model, as seen in Fig. 17. REMO2020<sub>40</sub> with iMOVE produces much more realistic LAI maps, with overestimations mainly in Fennoscandinavia (all seasons), Spain and Eastern Europe (summer), and Eastern Europe (autumn).

The harvest for crops in Europe typically occurs during the late summer months, depending on inter-annual temperature variability. The reduced LAI leads to reduced evapotranspiration of the vegetation and an increased role of the soil albedo, which is darker than the litter albedo in Europe (Rechid et al., 2009). On one hand, these processes increase the mean 2 m temperature and amplify the warm bias in the iMOVE simulations in the autumn season, particularly in Eastern Europe (Fig.

2), where cropland is one of the main land cover types in iMOVE. On the other hand, we also see reduced biases in the precipitation (Fig. A3). Moreover, we have already used REMO2020 with iMOVE with a newly developed irrigation module in (Asmus et al., 2023) and reported that the model, including vegetation, reacts very realistically to irrigation and provides a better representation of the local climate in irrigated areas. REMO2020 with iMOVE will be our main model for the land-use change simulations within the WCRP CORDEX FPS LUCAS project.

## 5 Conclusions







REMO2020 is the new version of the REMO REgional MOdel, representing the most significant update in the model's history. This new version is a major step towards a regional climate system model, as it integrates many previous physics-development versions, such as the lake model and vegetation model, into one unified system. REMO2020 also includes many new modules, such as time-varying aerosol climatologies and a multi-layer snow model, along with heavily updated and restructured physics packages. In terms of model dynamics, REMO2020 features a full non-hydrostatic extension and updated approaches for water advection in the atmosphere. This work focuses on the hydrostatic version of the REMO model, and the analysis has been conducted for the European CORDEX domain. The model has already been used in various projects in non-hydrostatic mode and for different domains.

This work not only introduces the new model version and its performance metrics but also consolidates all details of the soil module used in REMO into one publication.

REMO2015 is improved with REMO2020, especially when using 49 vertical levels, and the wintertime cold bias in Northern Europe is reduced, mainly due to the new multi-layer snow module. In some areas, however, like the Balkans, existing warm biases in autumn are enhanced with REMO2020. Precipitation biases are overall reduced in REMO2020 compared to REMO2015, but the model tends to overestimate orographic mountainous precipitation. This is linked to the higher vertical resolution used in the new model, which leads to grey-zone convective issues over mountainous regions, even with the 0.11° spatial resolution, as previously reported by Vergara-Temprado et al. (2020). Based on this earlier study, we tested the impact of deactivating deep convection, which improved precipitation patterns in some mountainous regions. It also, however, caused unrealistically high precipitation events and could not be used with REMO2020. Additionally, this work points out that REMO2015 already suffered from excessive extreme precipitation events, which we were able to improve by re-tuning the cloud schemes and other influencing factors from the updated model system.

We also analyzed how well the new model represents snow amounts in Northern Europe, and there were clear improvements compared to satellite measurement data. With higher vertical resolution, the results improved further, and activating the iMOVE module allowed us to improve the fractional snow cover, one of the new details added to REMO in this work. Improved snow representation also led to improvements in albedo representations. There are, however, still some remaining biases in REMO2020, likely linked to the simplified forest canopy approach, including the missing skin reservoir.

Cloud cover and vertical cloud fraction does not change significantly in the new version. While some of Northern Europe's positive cloud biases were removed with REMO2020, some underestimations near the Mediterranean area were enhanced, except in summertime. REMO2020, using 49 vertical levels, shows less underestimation but increases the positive cloud cover bias in the western parts of Europe. The vertical cloud fraction shows that the new version captures the overall features and gradients better than the old one but underestimates low-level cloudiness. Interestingly, although the new version was tuned to make it harder for the model to produce ice, REMO2020 still overestimates the ice water content more than REMO2015. This indicates that there is still room for improvement in the cloud scheme tuning parameters.

REMO2020 will be used for CMIP6 and CMIP7 Fast Track dynamical downscaling activities within the CORDEX project. Moreover, due to its modular structure REMO2020 is now well-suited for new development requirements arising from climate service needs, such as ongoing work with urban modeling.

Code and data availability. The sources for the REMO model are available on request from the Climate Service Center Germany (contact@ remo-rcm.de). Open access is not possible due to licensing limitations coming from the legacy code within REMO. The version used in this work is saved and achieved (Climate Service Center, 2025). All the scripts used to produce the results in this paper can be found from (Pietikäinen, 2025a). The model data is available from (Pietikäinen, 2025b). Section 3.2 provides a detailed description of all the measurement data used. Interested parties can refer to this section for information on how to download the data.

# Appendix A: Analysis plots

## A1 Temperature




Author contributions. J-PP, KS, LB, and TF led the model development, incorporating feedback from DJ. J-PP designed and conducted the simulation with assistance from LB. The main analysis was performed by J-PP, supported by KS and LB. CN contributed to the cloud analysis, while PH, CP, and DR focused on the vegetation analysis. J-PP authored the main sections of the manuscript, with DR, CP, and PH writing the land surface description summary. All authors reviewed the draft and contributed to the final paper.

Competing interests. The authors declare that they have no conflict of interest.

Acknowledgements. We want express our gratitude to Dr. Beiping Luo from ETH Zurich - Institute for Atmospheric and Climate Science, for preparing and providing the stratospheric aerosol forcing data for REMO. We acknowledge the E-OBS dataset from the EU-FP6 project UERRA (https://www.uerra.eu) and the Copernicus Climate Change Service, and the data providers in the ECA&D project (https://www.ecad.eu). The authors would like to acknowledge the work done by Dr. Thomas Raub with the prognostic precipitation before

**Figure A1.** Seasonal mean 2-m temperature minimums from E-OBS dataset and biases from different model versions. The seasonally averaged results are for the time period of 2001-2010.

Figure A2. Like Fig. A1, but for mean 2-m temperature maximums.

his departure to the Oldenburger OFFIS - Institut für Informatik. We want to thank the German Climate Computing Centre DKRZ for the high-performance computing capacity. We acknowledge that neither the European Commission nor ECMWF is responsible for any use that

**Figure A3.** Seasonal relative precipitation biases from different model versions against E-OBS data (See Fig. A1). Please note that areas with less precipitation than 0.1 mm/day in the multi-year seasonal sums have been excluded from the relative mean and RSME calculations. Still, some points with very little observed precipitation cause huge relative differences, which influence the mean and especially RSME.

may be made of the Copernicus information or data it contains. We would like to thank Microsoft Copilot for assistance with language polishing of this manuscript. We are grateful for the helpful suggestions by our colleague Dr. Claas Teichmann to improve the first draft of the manuscript. The authors would like to thank Dr. Sven Kotlarski from MeteoSwiss, Gabriela Juárez, and the anonymous referee for their valuable comments, which helped to improve the manuscript.

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
