# Peer review of "REMO2020: a modernized modular regional climate model"

_EGUsphere, 2025_

## Author Response (AR1)

**RC1: 'Useful and relevant paper on the most recent version of a widely used RCM, rather technical though', Sven Kotlarski, 25 May 2025**

Dear Dr. Kotlarski, we are very thankful for your review and constructive comments. Throughout the text your comments are marked with boldface and after each comment follows our reply.

The authors present a comprehensive description of their most recent edition of the REMO regional climate model (REMO2020). The new model version features a range of changes and extensions with respect to the original version REMO2015. These are described comprehensively and in detail. The manuscript also presents a thorough evaluation and comparison of several reanalysis-driven 10-year long simulations, using different model setups (e.g. two different numbers of vertical levels) and including a reference simulation of the old REMO2015 version. Results overall reveal an improved performance of REMO2020 with respect to its predecessor. The new version is planned to be employed for the upcoming CMIP6 and CMIP7 downscaling exercises in the frame of CORDEX.

Overall, the submission is well designed and fits into scope of the GMD journal. It is of general interest for the scientific community as it presents recent advances in a widely applied RCM. The manuscript has some touch of a technical documentation and is rather lengthy. The presentation and analysis of the 10-year long evaluation simulations however qualifies the submission to be presented as a scientific paper. The use of the English language is acceptable but could be improved in many places. I suggest additional editing by a native speaker. It is kind of difficult to read the entire paper en bloque, as many aspects of the new version are presented in depth. I believe there is some potential for an overall shortening and simplification of the manuscript, I'd however leave it up to the authors to implement additional changes in this respect. The selection and presentation style of the figures is appropriate. Despite the shortening suggestion, there is also one point that could/should be slightly extended: In the "code and data availability" it is mentioned that the source code is available from the author's institution, some more information would however be useful – in particular for a GMD submission. Is the entire code freely available and if so under which license? Are open source developments of the user community envisaged?

We acknowledge the technicality and lengthiness of the manuscript. In the next version, we will reduce the details for such implemented new features that can be referenced without losing critical information.

The code availability is still an issue for us due to legacy code limitations. We discussed this in detail with the Executive Editor and now the model version used is in Zenodo, but behind restricted access. The "code and data availability" part will be improved, and it will have information on why the sources cannot be publicly published and a reference to Zenodo. Overall, it is in our core values to follow the FAIR principles (findability, accessibility, interoperability, and reusability) and we have published all codes and data that we can.

Apart from these issues, I could generally recommend the manuscript for publication. Please find below a rather extensive list of further minor comments that should be considered before final publication. Congratulations to the authors for this nice piece of work!

**With kind regards. Sven Kotlarski**

Thank you. Also, many thanks for your constructive ideas on how to improve our manuscript!

**REMAINING MINOR ISSUES (additions/changes in CAPITAL letters):**

Model name: The number in the model name obviously refers to a year (the year of finalization of a specific model version?). I'm hence wondering why the new version is named "REMO2020" and not, for instance "REMO2025".

Not directly. The name REMO2020 was used already in previous publications and due to some delays, only now officially shown in detail. The update from previous versions was the biggest so far and nothing as major is planned in the near future. The name REMO2020 has a wider meaning, rather than pointing directly to the year it was published. This has been the case with different REMO versions before.

Lines 4-5: (i) THE Flake mode...(ii) A state-of-the-art ... (iii) A newly developed

Corrected as suggested (without capitalizing).

Line 7: "tuning" à l'd suggest to replace the term tuning by "calibration" throughout the entire paper. "Tuning" is partly understood as a non-scientific exercise, while "calibration" often represents a more thorough scientific exercise.

It is a good suggestion, but tuning is a widely used term, also used for calibration. We decided to continue using it.

Line 26: "including their future climate projections" could be skipped here.

That is true and it is removed.

Line 31: I believe "covering LAND AREAS OF the entire globe" would be more appropriate.

We changed it to "covering almost all land areas of the entire globe"

Line 37: "which PRODUCES more".

Corrected as suggested. Also changed the ending to "covering almost"

Line 43: "... to answer the FPS QUESTIONS ..."

Corrected as suggested.

Line 57: "... in the MODEL physics ..."

This part is rewritten.

Line 64: Better clarify: Will you introduce the existing REMO2015 version or the new REMO2020?

A good point. We made this part clearer.

Entire Chapter 2: It would be nice to occasionally point out which parts of the REMO features described are unique to REMO and which parts are nowadays "standard" in regional climate modelling (i.e.: relate REMO to other RCMs). This is however not a must and is completely left to the authors for deciding.

Also a good point. Sometimes it's a bit of a grey zone, but we tried to distinguish REMO specific parts, when possible.

Line 197: If the existing REMO2015 version indeed had errors that are fixed now, what does it mean for the use of existing REMO2015 simulations (e.g. those openly available through the ESGF)?

Based on our tests, these errors have an impact on the results, but they are not big enough to change the simulated climate. This means that REMO2015 results are still very usable. In REMO2020, these error sources are fixed and together with all other changes, improve the model's performance.

Line 220: "LW budget (outgoing component)" à the LW budget should include both incoming and outgoing LW, otherwise it would not be a budget. Please clarify.

This is true and we improved this part. Shortly, the incoming component is calculated on every radiation step (once per hour), but in the new version, the outgoing component is calculated on every time step. It would be ideal to update both at every time step, but due to the heavy radiation component, we can only afford to update the incoming component once per hour. The new approach does improve the tile-wise LW budget, although only the outgoing component is updated more frequently.

Lines 250ff: I wonder if the recalibration followed any specific scheme (such es e.g. presented by Bellprat et al. 2012, doi 10.1029/2012JD018262) or was more or less applied in random order.

No, we did not follow that or any other similar approaches directly, but the idea behind it had similarities. We had a detailed look at 10.1029/2012MS000154 (it is for ECHAM GCM, but that is the big-brother model for REMO and a good starting point) and tried to

understand as much as we could about different tuning parameters before changing anything. Many have a physical basis; they are not just a mystical number controlling some process. After this, targeted tests were made (looking at temperature, precipitation, cloudiness etc.) and when promising values or combinations of values were found, more detailed tests were made. We tried to avoid situations where a combination would give promising results, but one or more tuning parameters would have unrealistic values. By unrealistic we mean that if you then look at the specific process, it could give values that are not really correct but would improve some other areas by countering other inaccuracies. Indeed, this helped us to understand the model a bit more and, for example, when we had issues with the convection part, we straightaway knew how to look in the right place. Still, it would be also nice to test more sophisticated tuning methods in the future (combined with an understanding of different parameters).

**Line 266: "artificial top approach" à unclear.**

Indeed. It was supposed to link to the "REMO's land surface scheme" section, where the 10 cm artificial snow top layer approach was explained, but the linkage is almost non-existent. We will definitely improve this part and remind the reader about the artificial 10 cm top layer.

Line 277: "... Thus, the CHARACTERISTICS of ...".

Corrected as suggested.

Line 312: "... now UTILIZES updated ...".

This part has been rewritten.

Lines 335-336: Please specify the respective SWE unit (mm, cm).

Added unit [m].

Line 419: "the fraction of precipitation in a grid box" à unclear what is meant.

A good point. We have improved this part.

**Chapter 2.6: This chapter obviously refers to the prognostic nature of stratiform precipitation. What about convective precipitation then? If the latter is not considered a prognostic quantity: why not?**

An excellent point. The scheme we implemented cannot be used directly with our convective parameterization. The convection part remains as it is and could benefit from a new or modified scheme that would take the precipitation memory into account. Currently, the direct precipitation from the convective parameterization does not have a memory component, but the convective transported moisture will undergo the stratiform scheme and has indirectly the memory for precipitation. We will add more discussion about this to the manuscript.

**Line 436: "dampen the amplitude" à amplitude of what?**

We changed this to "dampen the solution amplitude in non-linear cases" Basically it means when solving any oscillations, in non-linear cases, like with climate, we might anymore get the amplitude (of the "solution") correctly (as compared to linear case where only mostly phase errors occur).

Line 480: You could slightly extend the introductory paragraph of Chapter 3 by better clarifying the purpose of the simulations. It is implicitly clear (comparison and evaluation of different setups and comparison to the previous REMO2015 version) but could be better motivated for the reader.

Again, a good point. We have extended the introduction paragraph.

Line 482: "SEVERAL REMO simulations ...".

Corrected as suggested.

Table 1: The systematics behind the different setups is not completely clear to me. Especially: the simulations setup does obviously not allow for a systematic comparison of 27 levels (always ERA-Int forcing) and 49 levels (always ERA5 forcing). Why is this the case and what are the consequences for the comparison of 27 vs. 49 levels?

We will add more explanation to this part (also a comment from Ref. 2). We wanted to have comparable simulation with 27 levels (ERA-Interim) with REMO2015 and REMO2020 and include the new default setup with 49-levels using the latest ERA5. This does mean, however, that some differences between REMO2020 27 levels and REMO2020 49 levels come from different lateral boundary forcing. Still, we did not see a necessity to rerun the 27 levels with ERA5 (with either version of REMO), because we are not going to produce such simulation anymore, not even for CORDEX.

**Line 506: This is a little unclear.**

Changed to "We did not include grid boxes with less than 21 days of data in a month when calculating the monthly averages for the analysis." This decision was made to filter out monthly values when they did not have enough data behind them.

Line 563: "changing" should be removed here.

That is true (removed).

Line 607: Replace by "WE ALSO SHOW THE 2-METER TEMPERATURE AND PRECIPITATION BIASES IN DIFFERENT...".

Corrected as suggested.

Lines 623-624: "differences are not that high", "perform really well" à such expressions should generally be avoided as the yare of extremely subjective nature. Wherever possible try to quantify the evaluation results.

This is true and we have improved the text.

**Line 627: Very unclear.**

This part has been totally re-written.

Figure 5: Zooming into the Vosges-Black Forest area, REMO2020 obviously considerably improves the long-known pattern of precip overestimation in the Upper

Rhine valley and underestimation in the hills of the Black Forest. This is a very welcome improvement. Is it related to the prognostic precipitation treatment in REMO2020?

We are also very happy with these results. It is a sum of many things. There are already improvements between REMO2015 and REMO2020 with 27-levels, which are coming from both the dynamical and physical improvements. Using the 49-levels also makes a big difference, but this step could not have been made without the prognostic scheme, so they go kind of together. It is impossible to point out one specific development step, but prognostic precipitation does play a big role.

Precipitation evaluation in general: Some positive precipitation bias could actually be expected due to observational undercatch (I believe EOBS has not been corrected for undercatch), especially in mountainous areas with a considerable snowfall fraction. This should at least be mentioned somewhere in the manuscript.

We were perhaps too cautious when discussing the measurement uncertainties (this also links to the Community Comment we got). This part will be improved in the manuscript.

Caption of Figures 7 and 8: I suggest to reword the caption to "DISTRIBUTION OF HOURLY JJA PRECIPITATION SUMS over Germany ..." to make it clearer (you do not show summer precipitation sums here).

Corrected as suggested.

Lines 661-662: "... as maximum values for ...".

Corrected as suggested.

Line 667: "... tendency for too INTENSE precipitation events ...".

Corrected as suggested.

Line 676: "grey-zone" needs to be better explained.

We have introduced an explanation for it.

Figure 10: I'm wondering why the SnowCCI dataset does not show any snow over the European Alps. Mean SWE sums >2 cm are surely present there. If I remember right, alpine regions were masked out in some SnowCCI product due to too large uncertainties in topographic terrain. But this should also be the case for the Scandinavian Alps then, which seem to have reasonable values. Could you comment on that?

The reason is that alpine regions are masked in the SnowCCI data. This also influences parts of the Scandinavian Mountains (as can be seen from Fig. 10; white constant areas in Western and Northwest Norway). We have improved the text to include information about the SnowCCI restrictions.

Line 876: "seamless" à I would avoid this expression here or at least better specify it. In the climate modelling context "seamless" typically refers to the time dimension, i.e. to integrating NWP, seasonal forecast, decadal forecasts and climate projections. This is obviously not eh case for REMO2020.

That is true. Changed to "unified"

The manuscript "REMO2020: a modernized modular regional climate model" by Pietikäinen et al. present a model description of new features in a revised version of the regional climate model REMO2015, called REMO2020. Model performance is analysed through ERA5 driven experiments in 12 km resolution over Europe for a 10-year period 2001-2010.

This paper would be useful as model documentation and validation for the REMO model, which is used extensively; but it would need some revisions. The paper is quite lengthy, and I don't think that all of the details of the algorithm description are relevant. In sections 2.1 and 2.2 among others, the very detailed descriptions do not seem to be related to updates, and would presumably already exist in earlier publications. I would like the authors to reduce the size of the model description with fewer details included; at the same time please brush up the language, which frequently seems a bit long-winded.

We have cleaned overall the manuscript and tried to leave only the changes we have made. In Section 2.2, however, we for the first time open all details related to REMO's surface treatment and have kept it as it is (2.1 is a short introduction of the model). The language was improved, as was requested also by Referee 1.

It is mentioned in p.17 that experiments with the two different vertical structures (27 and 49 levels) use different versions of ERA (ERA-Interim and ERA5). This prohibits any conclusions about the difference between the two model setups. I urge the authors to complete the set of experiments with the more recent ERA5 reanalysis as boundary conditions for all REMO configurations.

Here we wanted to keep the existing REMO2015 configuration, as the data it produced has been used widely, also in CORDEX. To show some main differences between the model versions, we used REMO2020 with 27-levels with ERA-Interim. The new default configuration, with 49-levels, used the latest ERA5, to show the current performance. It is true that some of the direct comparison we show suffers from the different forcing data, but we do not see the need to rerun the 27-level version with ERA5 or use ERA-Interim for the 49-level version. Neither of these will be used in the future and would not bring

| anything new from that perspective, although that would allow for more direct comparisons.                                                                                                              |
|---------------------------------------------------------------------------------------------------------------------------------------------------------------------------------------------------------|
| Minor things:                                                                                                                                                                                           |
| l248 please reformulate (non-English phrasing).                                                                                                                                                         |
| Reformulated.                                                                                                                                                                                           |
| Please show relative change precipitation change in Fig. 3 and put absolute changes in the Appendix instead of the converse. This is the most common way to illustrate climate change of precipitation. |
| Perhaps more common, but both are used. We find the current approach to be more informative for the analysis.                                                                                           |
| l627 Remove "is"                                                                                                                                                                                        |
| Removed.                                                                                                                                                                                                |
| l644 resolve "in from"                                                                                                                                                                                  |
| Resolved (from).                                                                                                                                                                                        |
| Section 4.2.4 ought to be called something like "Precipitation intensity spectra", as "precipitation distribution" will be read as spatial distribution.                                                |
| Changed to "Precipitation probability distribution"                                                                                                                                                     |
| l662have higher intensity events should be reformulated. May behave a higher number of high-intensity events  Corrected as suggested.                                                                   |

**CC1: 'Limitations of Regional Climate Models in Representing Precipitation: Implications for Model Use and Development', Gabriela Juárez, 03 Jun 2025**

We thank you for this community comment. Throughout the text your comments are marked with boldface and after each comment follows our reply. Overall, we would have appreciated a more scientific approach to criticisms, pointing out in details how did you drew your conclusions and referencing previous studies to back up your claims. This would help us to understand which of your claims are scientific criticisms and which are opinions, eventually improving our manuscript.

The results presented in this paper highlight significant and ongoing shortcomings in precipitation modelling using RCMs. This is evident in Figures 3 and A3. Errors exceeding ±100% compared with observed precipitation across wide areas highlight the difficulty of producing realistic precipitation fields in both mountainous and flat regions.

We have values over 100% and the manuscript discusses in detail the reasons for mountainous areas. For flat regions with higher biases, one has to also take into account the seasonality and how much precipitation there is. During dry seasons, small changes in absolute values (Fig. 3) can lead to higher changes in relative values (Fig. A3), which is the case with many flatter areas with high relative bias. The new model version improves the relative bias in flat areas considerably when compared to the old version, a point that we will make clearer in the next version of the manuscript. Thank you for your comment, it points out that this part was not clear enough before.

The figures for precipitation demonstrate poor overall performance. Indeed, precipitation is a critical variable for evaluating climate models. As the 'final' product of modelling, precipitation is subject to the cumulative effect of errors in thermodynamics and dynamics. In order to correctly simulate precipitation, it is first necessary to successfully model longwave and shortwave radiation, the onset and strength of convection, humidity, and the microphysics of liquid, solid, and mixed phases with a certain degree of precision. It is also necessary to model the dynamics of the atmosphere well so that air density, pressure, wind and temperature are in the

right place at the right time. This makes precipitation a valuable metric with which to evaluate model performance.

The new version has better metrics for precipitation. The figures show clear improvement, from a model that was performing well in previous multi-model evaluation studied (see Kotlarski et al. (2014)). The authors cannot agree on the claim of poor overall performance; this paper clearly shows this is not the case. We welcome you to comment on the new forthcoming EURO-CORDEX CMIP6 downscaling evaluations. These should be available before the end of the year.

Despite advances in microphysics and how convection is treated in the new model, the results suggest that this improved version of the RCM still does not adequately capture the underlying physics. The community knows that modelling is complicated, so that is not a problem as long as it is fully acknowledged in the paper. Otherwise, the reader would be misled.

We have quite openly shown the model performance in many different variables. It is unclear to us how did you came up with your conclusion.

These limitations should prompt a more cautious narrative about the model's predictive ability than is conveyed in the introduction. Rather than presenting the RCM as a robust tool for downscaling, the authors should acknowledge its limitations more explicitly. Their enthusiasm for high-resolution output should be balanced with a more nuanced assessment of model fidelity, given the results presented. The enthusiastic assessment of performance is inconsistent with the actual performance shown for the present climate. The discrepancies cast serious doubts on the model's ability to capture the current climate, which is amplified for future climates and hinders its ability to provide societal advice.

We openly show how the new model version is better than the old one without hiding any biases or issues we encountered during development or analysis. We are talking here about a regional climate model, and as you mention yourself later, no model is perfect, and this is a known thing. It would be crucial for our reply to know what exactly you mean by the discrepancies.

A note on recent developments in AI and cloud-resolving Earth System Models would also be needed. It is clear to the reader that these developments challenge the continued investment in RCMs, particularly given their difficulty in modelling precipitation, the basic metric. While the rationale for refining an RCM may still hold in certain research contexts, this must be clearly articulated — especially in light of long-standing criticisms such as those made by Trenberth (2007) regarding their suitability for modelling so-called 'Mediterranean hurricanes'.

It is impossible to draw a conclusion about RCMs difficulty to model precipitation based on this work, or the work you referenced to (we assume you are talking about <a href="https://doi.org/10.1029/2006JD008304">https://doi.org/10.1029/2006JD008303</a>). These studies, if correctly assumed, use only one forecast model and cannot be generalized. Even further, they show the importance of properly modelling the surface energy fluxes, an area that we improved within our work. In a broader context, yes AI and steps in global modelling are important factors, but should they be explicitly mentioned in a regional model development paper is another issue. We do mention in our introduction that even with coarser resolution, long transient climate simulations are challenging for GCMs, and still quite impossible for cloud-resolving scales. These simulations, however, are needed when looking at the climate and climate change impacts, highlighting the need for RCM simulations. The time will come when GCMs are able to produce more high-resolution long transient simulations, but even at this stage, RCMs might play a big role in simulating different adaptation strategies, for example.

The simplifications in the microphysics and convection schemes of this RCM also call into question the rationale behind producing detailed precipitation fields at a resolution of 0.11° under such assumptions. A major motivation for RCM modelling is to increase spatial resolution compared to GCM/ESM models, many of which now have more sophisticated schemes than RCMs. A comment on this would benefit the paper.

Some RCMs, like REMO, have their cloud scheme roots in global climate models. Others use numerical weather prediction (NWP) based approaches in their cloud schemes. As resolution increases, especially to non-hydrostatic scales, the underlaying assumptions in the cloud schemes should be re-checked. In our case, we implemented the prognostic precipitation to improve the "precipitation memory" issue. It would be interesting to know where the claim that GCMs have more sophisticated schemes arises? We should also not

forget that motivation for RCMs is not only increasing spatial resolution, but also better representation of the underlaying surface, which plays a big role.

On the same line, as our modelling capacities have evolved greatly over the past 30 years, so should our criteria for allocating computational and research resources. Should these resources be used to patch up legacy RCMs or to develop more advanced alternatives?

Building a new model, or even something based on existing, can be really time- and resource-consuming work and still have updated legacy components. Updating legacy RCMs (or GCMs or any other legacy model), depends on the model skill and technical challenges arising from the supercomputing developments. In our work, we have shown that a major update on REMO, even if it still has legacy code in it, has been beneficial, has improved climate simulations, and will be used in future activities including CORDEX.

The fact that even sophisticated microphysical schemes like P3 do not substantially improve performance suggests that there are deeper structural limitations in our knowledge of precipitation. This raises a broader issue: how do CORDEX and dynamic downscaling approaches compare with global models? If the added resolution does not result in greater realism, the value of the framework must be questioned. The results show that the modelled fields do not compare well with observations at the native resolution.

Again, we do not agree with your conclusions, and the manuscript shows this. We would appreciate a more detailed rationale behind your claims if you wanted to draw such conclusions. The new CMIP6 CORDEX downscaled results are soon available, and evaluation, including comparison to GCMs, will be available. We welcome presenting your views in these papers; here, regardless of the skill, one model does not represent the whole ensemble.

Cloud parameterisation is another critical area that requires scrutiny in the paper. While acknowledging variability in cloud droplet concentration with height and geography, REMO2020's representation of stratiform and convective clouds lacks a quantitative sensitivity analysis. Without an understanding of how the assumptions may affect precipitation and temperature outputs, any conclusions drawn from the model must be treated with caution. It is necessary to provide a note on which

empirical values and assumptions are used in the microphysics and in modelling convection.

Both schemes have their assumptions, parameters, tuning variables etc. We have all references to the original descriptions of the schemes, and the texts explain the changes made. If there is something specific that you think should be shown here, we would be happy to hear the reasons why and further discuss it. We got feedback from the 2 reviewers that already now we have too many technical details in the manuscript, although we mainly included details of the newly implemented features. Overall, everything that RCMs/GCMs/model have inside cannot be opened in every single publication (that is why we use references and cite the corresponding sources).

On a positive note, unlike in other papers, the authors do not try to hide the limitations by upscaling the fields or using probability density functions to make comparisons. While the manuscript's focus on native model resolution is commendable, it requires a more nuanced approach on some technical issues arising from that choice. While high resolution offers the potential to capture small-scale processes, it also increases susceptibility to noise and spurious features. A clear discussion is needed on how REMO2020 addresses artefacts.

We do have probability density functions in our paper. We also discuss some limitations coming from the model side and from the measurement side. Overall, we will improve this part of the manuscript.

In summary, this study highlights significant limitations in the RCM framework. Rather than downplaying these issues, the authors should openly examine them to improve model transparency and ensure appropriate use outside the scientific realm. As it stands, the paper creates a misleading impression of the actual performance of RCMs and the current state of precipitation science.

First of all, we still disagree with your conclusions and have shown in the paper why. Secondly, we are more than happy to take critic, but critic with explanations. Finally, whatever one model's skill is, how can you draw a conclusion about all RCMs based on that?

While RCMs can still play a valuable role in process studies, such as hurricane intensification, their use in generating regionalised climate scenarios requires greater caution than is often applied in the narrative. This paper should not perpetuate the poor practice of concealing the limitations of the dynamical downscaling approach from the public.

On the contrary, we even list many successful projects which are based on RCM data. Using RCM ensemble data, for example through CORDEX, is the backbone for almost all climate services in Europe, and the skill has been shown in numerous studies from better spatial representation of precipitation fields to much better representation of the precipitation density function extremes. We welcome you to familiarize yourself with them (references are in our introduction).

**References:**

Kotlarski, S., Keuler, K., Christensen, O. B., Colette, A., Déqué, M., Gobiet, A., Goergen, K., Jacob, D., Lüthi, D., van Meijgaard, E., Nikulin, G., Schär, C., Teichmann, C., Vautard, R., Warrach-Sagi, K., and Wulfmeyer, V.: Regional climate modeling on European scales: a joint standard evaluation of the EURO-CORDEX RCM ensemble, Geosci. Model Dev., 7, 1297–1333, https://doi.org/10.5194/gmd-7-1297-2014, 2014.